# Modality Curation: Building Universal Embeddings for Advanced Multimodal Information Retrieval

## Abstract

Multimodal information retrieval (MIR) faces inherent challenges due to the heterogeneity of data sources and the complexity of cross-modal alignment. While previous studies have identified modal gaps in feature spaces, a systematic approach to address these challenges remains unexplored. In this work, we introduce UNITE, a universal framework that tackles these challenges through two critical yet underexplored aspects: *data curation* and *modality-aware training configurations*. Our work provides the first comprehensive analysis of how modality-specific data properties influence downstream task performance across diverse scenarios. Moreover, we propose Modal-Aware Masked Contrastive Learning (MAMCL) to mitigate the competitive relationships among the instances of different modalities. Our framework achieves state-of-the-art results on multiple multimodal retrieval benchmarks, outperforming existing methods by notable margins. Through extensive experiments, we demonstrate that strategic modality curation and tailored training protocols are pivotal for robust cross-modal representation learning. This work not only advances MIR performance but also provides a foundational blueprint for future research in multimodal systems. Our project is available at https://anonymous.4open.science/r/UNITE-Embedder-09BC.

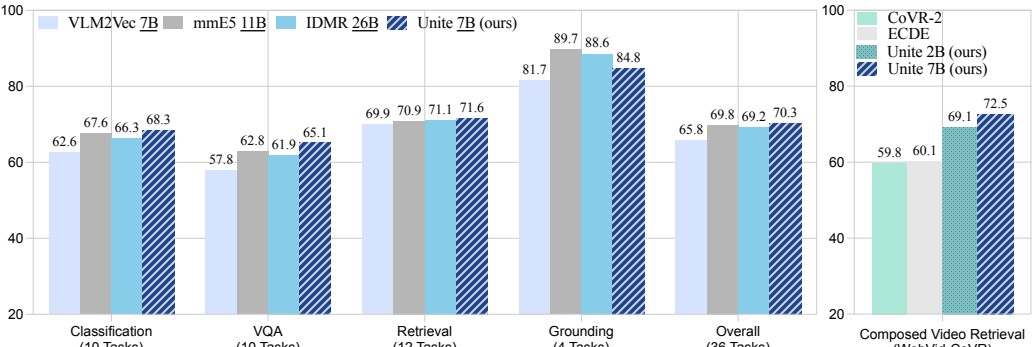

Figure 1: Performance comparison on instruction-based retrieval benchmarks (left: MMEB (Jiang et al., 2024c) and right: WebVid-CoVR (Ventura et al., 2024a)). Our UNITE achieves leading performance on various tasks, even surpassing models with larger parameter scales.

## 1 Introduction

Multimodal Information Retrieval (MIR) is a critical research topic (Radford et al., 2021; Jia et al., 2021), aiming to satisfy users' information requirements for diverse media, such as text, images, and videos. As multimedia applications continue to progress and develop, a series of more complex and demanding tasks come to the fore, which are collectively referred to as fused-modal retrieval such as the retrieval of composite images or videos (Liu et al., 2021; Zhang et al., 2024c; Ventura et al., 2024a). These tasks require highly sophisticated approaches for handling interleaved multimodal queries and candidates, highlighting the necessity for the development of a one-piece framework for unified multimodal representations.

Recently, large multimodal models (LMMs) have shown powerful capabilities on various vision-language tasks, such as visual question answering (VQA) (Antol et al., 2015; Kafle & Kanan, 2017; Wang et al., 2017; Marino et al., 2019) and multimodal fact-checking (Yao et al., 2023; Akhtar et al., 2023). In MIR field, several methods (Jiang et al., 2024c; Liu et al., 2024d; Gu et al., 2025) have explored adapting large language models (LLMs) for retrieval tasks via contrastive learning, aiming to produce unified embeddings. For example, E5-V (Jiang et al., 2024b) finetunes LLaVA-NeXT (Liu et al., 2024a) with text-only NLI (Gao et al., 2021) data, demonstrating the portability of LMMs for multimodal retrieval. GME (Zhang et al., 2024d) achieves leading performance in various image-text retrieval by finetuning Qwen2-VL (Wang et al., 2024b) on diverse image-text datasets. InternVideo2 (Wang et al., 2024c) stands out prominently in text-video retrieval, due to its training process that involves on hundred millions of video-text pairs. Although achieving notable success in specific domains, these models are hindered by their limited modalities, which inherently restrict their ability to fully capitalize on the potential of LMMs for generating unified multimodal embeddings.

Despite ongoing researches (McKinzie et al., 2024; Zhang et al., 2024b; Zohar et al., 2024) exploring training strategies for LMMs in MIR–including model architectures, training methodologies, and dataset considerations–critical questions remain unresolved. Specifically, the optimal data composition and proportions, as well as the nuanced impact of different modal data configurations across various retrieval tasks, have yet to be comprehensively understood. In our empirical investigations, we find that inappropriate combinations of multimodal data or data training sequences can easily disrupt the harmonious integration of diverse data modalities, causing the model to misinterpret the relationships between different types of information.

In this paper, through a meticulous analysis of how various data compositions impact retrieval results, we make efforts to achieve a balance among the three modalities of text, image, and video. In particular, we find that introducing a small number of fine-grained video-text pairs during the retrieval adaptation stage can significantly enhance the fine-grained retrieval performance of LMMs. Meanwhile, existing works (Jiang et al., 2024b; Zhang et al., 2024d) have proven that there is a substantial distribution gap in the data of various modalities within the feature space. Simply mixing data from different modalities for contrastive learning will affect the quality of representation learning (*i.e.*, introducing noise). Therefore, we propose *Modal-Aware Masked Contrastive Learning* (MAMCL) to balance the competitive relationships among the instances of various modalities.

Finally, we develop a **UNI**versal mul**T**imodal **E**mbedder, named UNITE, that effectively handles text, images, videos, and their combinations. During the training, we employ an evolving training strategy to gradually unlock the retrieval capability of LMMs. After two training stages (*i.e.*, retrieval adaptation and instruction tuning), we validate our approach through comprehensive evaluation on 40+ diverse retrieval tasks, spanning coarse-grained, fine-grained, and instruction-based retrieval across text, images, and video. Experimental results show that UNITE achieves state-of-the-art performance in various tasks, and outperforms existing specialized domain-specific models in numerous scenarios. In summary, our contributions are as follows:

- We unveil the appropriate method for curating modality data during the process of learning unified multimodal embeddings, so as to balance the gap between the feature spaces of different modalities.
- We propose MAMCL to mitigate the competitive relationships among the instances of various modalities. Specially, the MAMCL strategy can serve as a general method and be applied to any extended modal scenarios.
- To the best of our knowledge, UNITE is the first model capable of enabling text, image, video and fused-modal to concurrently focus on fine-grained and instruction-based retrieval tasks. Notably, UNITE achieves state-of-the-art performance in 40+ different tasks, surpassing specialized domain-specific models in numerous scenarios, as shown in Figure 1.

## 2 RELATED WORK

**Large Multimodal Models.** LMMs have mushroomed, showcasing impressive capabilities in multimodal information understanding (Li et al., 2023; Zhang et al., 2025). Pioneering efforts such as LLaVA (Liu et al., 2024b), MiniGPT-4 (Zhu et al., 2023c), InternVL (Chen et al., 2024b), and Qwen-VL (Wang et al., 2024b), bridge visual encoders and LLM with lightweight intermediate architectures. Recent advances (Li et al., 2024; Wang et al., 2025b; Zohar et al., 2024; Xu et al., 2024a) have extended these techniques from static images to sequential videos, demonstrating promising results

in video understanding through frame-based processing. These developments have catalyzed the widespread adoption of LMMs across various applications (Liu et al., 2024c; Pan et al., 2023).

**Cross-Modal Retrieval.** Traditional multimodal retrieval primarily focuses on cross-modal scenarios, particularly text-image retrieval (Wang et al., 2016; 2025a) and text-video retrieval (Zhu et al., 2023b; Jiang et al., 2022). Foundational methods, such as CLIP (Radford et al., 2021), Align (Jia et al., 2021), BLIP (Li et al., 2022) and CoCa (Yu et al., 2022), separately encode text and images through dual-encoder structures, and learn multimodal representations by contrastive learning on large-scale image-text pairs. ImageBind (Girdhar et al., 2023), OmniBind (Wang et al., 2024d) and VAST (Chen et al., 2023) expand this approach to accommodate more modalities with similar architectures.

**Instruction-based Retrieval.** Recently, the community has witnessed growing demand for multi-modal information retrieval in complex scenarios, including composed image/video retrieval (Liu et al., 2021; Saito et al., 2023; Hummel et al., 2024), multimodal document retrieval (Mathew et al., 2021; Dong et al., 2025), and multimodal knowledge retrieval (Luo et al., 2023; Long et al., 2024). While CLIP-based models struggle with complex multimodal queries, LMMs naturally excel at processing fused-modal inputs within a universal framework. Recent methods (Zhang et al., 2024d; Lin et al., 2024; Lan et al., 2025) contrastively train LMMs (*e.g.*, Qwen2-VL and LLaVA-NeXT) to uniformly embed images and text, boosting complex fused-modal retrieval. However, these models still face limitations in video-related tasks compared to video-specialized methods, highlighting the need for a universal multimodal embedder that excels across text, image, video, and their combinations.

## 3 UNITE

### 3.1 TASK FORMULATION

Unified multimodal retrieval addresses queries and candidates across text, image, and their combinations (Wei et al., 2024; Zhang et al., 2024d). We extend prior formulations to include video, enabling more comprehensive multimodal embedding alignment. We define a query $\mathbf{q} \in \mathcal{Q} \subset \mathbb{R}^d$ representing an embedding that is sampled from a specific modality. Specifically, $\mathbf{q}_t$, $\mathbf{q}_i$ and $\mathbf{q}_v$ denote the text, image and video embeddings, respectively. In practice, the query can be any single modality or a combination of various modalities (*e.g.*, $(\mathbf{q}_t, \mathbf{q}_i)$, $(\mathbf{q}_v, \mathbf{q}_t)$). Similarly, we define a retrieval candidate $\mathbf{c} \in \mathcal{Q} \subset \mathbb{R}^d$ presenting the representation embeddings of a specific modality. Thus, $\mathbf{c}_t$, $\mathbf{c}_i$ and $\mathbf{c}_v$ denote the text, image and video embeddings, respectively. The main purpose of MIR is to maximize the correlation between the related query $\mathbf{q}$ and candidate $\mathbf{c}$ pairs in the repsentation space $\mathcal{Q}$, while ensuring that the correlation between the unrelated query $\mathbf{q}$ and candidate $\mathbf{c}$ pairs is as low as possible.

In practice, for each query $\mathbf{q}$, we can collect its corresponding positive candidate $\mathbf{c}^+$ and a set of negative candidates $\mathcal{C}^- = \{\mathbf{c}_1^-, \ldots, \mathbf{c}_K^-\}$, where $\mathcal{C}^-$ consists of both in-batch negatives and hard negatives, $K$ refers to the number of negative candidates. As aforementioned, both queries and candidates can encompass various modalities. We employ contrastive learning with the InfoNCE loss (Oord et al., 2018) to simultaneously minimize the distance between matched query-candidate pairs $(\mathbf{q}, \mathbf{c}^+)$ while maximizing the distance between unmatched pairs $(\mathbf{q}, \mathbf{c}^-)$:

$$\mathcal{L}_{\text{CL}} = -\log \frac{\exp(\cos(\mathbf{q}, \mathbf{c}^+)/\tau)}{\exp(\cos(\mathbf{q}, \mathbf{c}^+)/\tau) + \sum_{\mathbf{c}_i^- \in \mathcal{C}^-} \exp(\cos(\mathbf{q}, \mathbf{c}^-)/\tau)} \quad (1)$$

where $\tau$ is the temperature hyper-parameter.

### 3.2 MODEL ARCHITECTURE

Generally, LMMs are composed of three essential components: a LLM, a vision encoder, and a vision projector, as shown in Figure 2 (a). This architectural design empowers LMMs to handle text, images, videos, and their integrated forms with remarkable fluidity and efficiency. This native multimodal processing provides new promise for unified multimodal embeddings. Following recent methods (Jiang et al., 2024c; Zhang et al., 2024d), we extract target embeddings from the hidden state of the final token in the last layer. Inspired by PromptEoL (Jiang et al., 2024a) and E5-V (Jiang et al., 2024b), we adapt the Explicit One word Limitation (EoL). Specifically, we use prompt template:

```
<vision>\n<text>\nSummarize above <modalities> in one word:
```

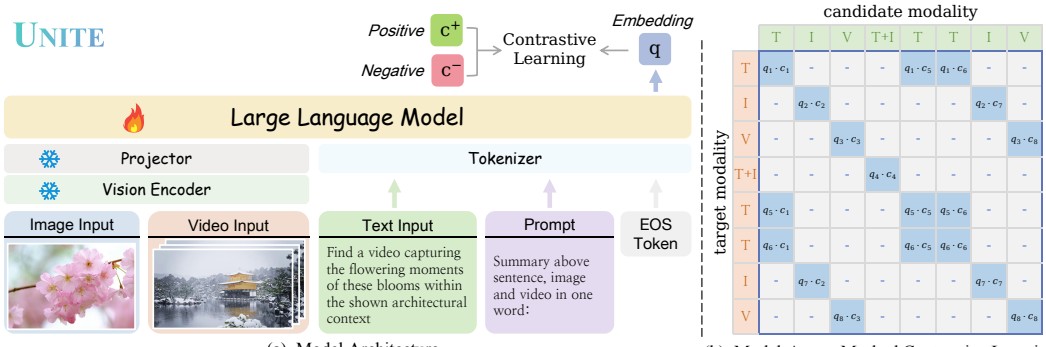

(a) Model Architecture

(b) Modal-Aware Masked Contrastive Learning

Figure 2: Overview of UNITE: (a) Model architecture utilizing LMM as the backbone, supporting multimodal inputs (text, images, videos, and their combinations). (b) Similarity matrix after applying MAMCL, which enables focused contrastive learning by restricting comparisons to samples sharing the same target modality, thus reducing inter-modal interference.

where `<vision>` and `<text>` are placeholders for visual content (*i.e.*, images, videos) and textual sentences, respectively, and `<modalities>` specifies the input modality types.

## 3.3 TRAINING STRATEGY

We employ a two-stage training scheme: retrieval adaptation and instruction tuning. In the first stage, we adapt LMMs to general retrieval tasks, while we enhance the capabilities of LMMs for instruction-based complex retrieval scenarios in the second stage.

**Retrieval Adaptation.** LMMs have showcased remarkable prowess in multimodal understanding, adeptly handling and interpreting diverse forms of information. However, retrieval tasks remain uncharted territory for these models. Thus, in the first stage, we focus on building robust fundamental retrieval capabilities by exposing LMMs to various information scenarios. It enables the LMMs to learn and adapt to the varying characteristics and requirements of different retrieval tasks. For example, we utilize single-modal and multimodal retrieval data, including text-text, image-text and video-text pairs, in the experiments.

**Instruction Tuning.** To enhance generalization across varied multimodal retrieval scenarios, we employ instruction tuning using comprehensive datasets like MMEB (Jiang et al., 2024c), which covers 36 datasets across 4 multimodal tasks. This stage introduces complex fused-modal retrieval scenarios via instruction-guided samples. This goes beyond the basic single-modal and multimodal retrieval tasks, enabling a more sophisticated and nuanced understanding of retrieval tasks.

## 3.4 MODAL-AWARE MASKED CONTRASTIVE LEARNING

Traditional multimodal retrieval models typically employ standard InfoNCE loss (Oord et al., 2018) (*i.e.*, Eq. (1)) in the contrastive learning, it treats all negative pairs equally regardless of the modality compositions. However, this strategy overlooks the inherent distinctiveness of diverse modal combinations in retrieval tasks. For instance, embeddings derived solely from text and those from multimodal sources typically display substantial disparities in their distributions within the feature space. When subjected to joint contrastive learning, the model struggles to balance the diverse information from different modalities, resulting in representations that fail to fully capture the semantic richness of each modality. This leads to our hypothesis: *Contrastive learning performed across instances featuring disparate target modalities has the potential to introduce noise and give rise to negative effects.*

To address this crucial problem, we propose *Modal-Aware Masked Contrastive Learning* (MAMCL), introducing a modality-aware constraint to mitigate competitive relationships between various target modal instances. Given a batch of $N$ samples, we compute the similarity matrix $\mathbf{S} \in \mathbb{R}^{N \times (1+K)}$, where $1 + K$ refers to the number of all positive and negative candidates, $\mathbf{S}_{nk}$ represents the cosine similarity between query $\mathbf{q}_n$ and candidate $\mathbf{c}_k$:

$$\mathbf{S}_{nk} = \cos(\mathbf{q}_n, \mathbf{c}_{nk})/\tau \tag{2}$$

Table 1: Zero-shot performance of fine-grained video-text retrieval on CaReBench (Xu et al., 2024b). LLaVA-NV refers to LLaVA-NeXT-Video (Zhang et al., 2024e). † indicates that these models have been equipped with feature representation capabilities through contrastive learning. *Rel.*Δ represents the relative performance improvement of UNITE$_{base}$ 7B compared to UNITE$_{base}$ 2B. Results in **bold** and underline denote the best and second-best performances.

| Model | CaRe-General | | | | CaRe-Spatial | | | | CaRe-Temporal | | | |
|---|---|---|---|---|---|---|---|---|---|---|---|---|
| | T→V | | V→T | | T→V | | V→T | | T→V | | V→T | |
| | R@1 | R@5 | R@1 | R@5 | R@1 | R@5 | R@1 | R@5 | R@1 | R@5 | R@1 | R@5 |
| *CLIP-based Models* | | | | | | | | | | | | |
| CLIP L/14 | 51.2 | 83.4 | 54.7 | 86.9 | 49.0 | 81.9 | 55.4 | 85.6 | 33.5 | 70.3 | 39.7 | 76.2 |
| LanguageBind | 64.3 | 91.0 | 59.5 | 88.0 | 64.7 | 90.8 | 61.0 | 87.2 | 39.8 | 77.3 | 42.2 | 77.6 |
| Long-CLIP L/14 | 62.7 | 88.8 | 60.3 | 88.8 | 65.6 | 90.9 | 61.0 | 88.3 | 33.2 | 68.8 | 34.5 | 71.9 |
| InternVideo2$_{stage2}$ 1B | 72.5 | 93.7 | 69.5 | 94.6 | 72.4 | 94.2 | 62.7 | 90.5 | 46.0 | 80.8 | 46.6 | 82.5 |
| *LMM-based Models* | | | | | | | | | | | | |
| LLaVA-NV 7B† | 66.9 | 89.4 | 62.7 | 89.2 | 68.0 | 92.0 | 65.0 | 90.0 | 43.3 | 76.9 | 40.1 | 75.4 |
| MiniCPM-V 2.6† | 71.0 | 92.2 | 69.3 | 92.8 | 71.7 | 93.6 | 67.6 | 92.3 | 50.5 | 82.9 | 46.1 | 80.9 |
| InternVL2 8B† | 72.1 | 92.6 | 73.6 | 93.4 | 76.1 | 94.1 | 74.3 | 94.5 | 48.1 | 76.8 | 47.6 | 78.2 |
| Tarsier 7B† | 71.0 | 93.8 | 70.6 | 94.2 | 70.2 | 94.0 | 67.4 | 93.5 | 50.1 | 84.1 | 50.0 | 84.7 |
| Qwen2-VL 7B† | 76.6 | 95.3 | 77.4 | 95.6 | 78.2 | 95.5 | 75.4 | 95.0 | 51.9 | 84.8 | 52.7 | 85.4 |
| CaRe 7B | 77.0 | 95.6 | 79.0 | 96.8 | 76.8 | 96.3 | 78.1 | 95.8 | 50.7 | 85.3 | 53.4 | 86.3 |
| UNITE$_{base}$ 2B | 78.1 | 95.5 | 80.8 | 96.4 | 79.6 | 95.4 | 78.0 | 95.4 | 45.3 | 77.6 | 50.0 | 83.6 |
| UNITE$_{base}$ 7B | 86.0 | 96.8 | 86.9 | 98.3 | 86.5 | 96.9 | 84.8 | 98.0 | 52.4 | 82.5 | 55.4 | 86.5 |
| *Rel.*Δ | +10.1% | +1.4% | +7.5% | +2.0% | +8.7% | +1.6% | +8.7% | +1.6% | +15.7% | +6.3% | +10.8% | +3.5% |

where $n \in \{1, \ldots, N\}$ and $k \in \{1, \ldots, 1+K\}$, $\mathbf{c}_{nk}$ refers to the $k$-th candidate of the $n$-th query sample. To incorporate modality awareness, we introduce a modality mask matrix $\boldsymbol{\mathcal{M}} \in \{0,1\}^{N \times (1+K)}$, where $\boldsymbol{\mathcal{M}}_{nk}$ indicates whether candidate $\mathbf{c}_{nk}$ shares the same target modality combination as the positive candidate $\mathbf{c}_n^+$ of query $\mathbf{q}_n$:

$$\boldsymbol{\mathcal{M}}_{nk} = \mathbb{1}[\Phi(\mathbf{c}_n^+) = \Phi(\mathbf{c}_{nk})] \tag{3}$$

where $\Phi(\cdot)$ refers to the operation of extracting the modality type of the embeddings. This extraction can be directly accomplished using the prior knowledge of the inputs. In this manner, we ensure that each query only considers candidates with the same modality as its target candidate during the contrastive learning. Then, we update the masked similarity matrix $\tilde{\mathbf{S}}$ as follows:

$$\tilde{\mathbf{S}}_{nk} = \mathbf{S}_{nk} \cdot \boldsymbol{\mathcal{M}}_{nk} + (-\infty) \cdot (1 - \boldsymbol{\mathcal{M}}_{nk}) \tag{4}$$

A visualization of the similarity matrix $\tilde{\mathbf{S}}$ is shown in Figure 2 (b). Finally, we expand the Eq. (1) to incorporate negatives, constantly ensuring that the model remains sensitive to different modalities:

$$\mathcal{L}_{\text{MAMCL}} = -\frac{1}{N} \sum_{n=1}^{N} \log \frac{\exp(\tilde{\mathbf{S}}_{np})}{\exp(\tilde{\mathbf{S}}_{np}) + \sum_{k=1, k \neq p}^{K+1} \exp(\tilde{\mathbf{S}}_{nk})} \tag{5}$$

where $p$ refers to the index of the positive candidate.

## 4 EXPERIMENTS

**Experimental Setup** For the retrieval adaptation stage, we curate a diverse 7M-instance dataset spanning four categories: (1) text-text pairs from MSMARCO (Bajaj et al., 2016), NLI (Gao et al., 2021), NQ (Kwiatkowski et al., 2019), HotpotQA (Yang et al., 2018), Fever (Thorne et al., 2018), TriviaQA (Joshi et al., 2017) and SQuAD (Rajpurkar et al., 2016); (2) image-text pairs from CapsFusion (Yu et al., 2024), LAION-Art (Schuhmann et al., 2022) and MSCOCO (Lin et al., 2014); (3) video-text pairs from InternVid-10M-FLT (Wang et al., 2023); (4) fine-grained video-caption pairs Tarsier2-Recap-585K (Yuan et al., 2025). We name the model trained in this stage as UNITE$_{base}$. For the instruction tuning, we use the combination of MMEB (Lin et al., 2024) and WebVid-CoVR (Ventura et al., 2024b) as our training set. Thus, we name the model trained in this stage as UNITE$_{instruct}$. We will explain the underlying rationale for this data composition in Section 5. Our specific training data composition and allocation are shown in Appendix A.1. We used Qwen2-VL (Wang et al., 2024b) as the backbone of our model to conduct experiments on

Table 2: Results of fine-grained image-text retrieval on ShareGPT4V, Urban1K, and DOCCI.

| Model | ShareGPT4V | | Urban1K | | DOCCI | |
|---|---|---|---|---|---|---|
| | T→I | I→T | T→I | I→T | T→I | I→T |
| *CLIP-based Models* | | | | | | |
| CLIP L/14 | 83.6 | 84.2 | 55.6 | 68.3 | 65.8 | 63.1 |
| OpenCLIP L/14 | 81.8 | 84.0 | 47.0 | 47.0 | - | - |
| Long-CLIP L/14 | **97.3** | **97.2** | 86.1 | 82.7 | 78.6 | 66.5 |
| EVA-CLIP 8B | 93.1 | 91.2 | 80.4 | 77.8 | - | - |
| FineLIP | - | - | 94.1 | 93.2 | 86.0 | 84.5 |
| *LMM-based Models* | | | | | | |
| MATE | - | - | - | - | 84.6 | 76.6 |
| E5-V 7B | 85.1 | 82.1 | 88.9 | 83.2 | - | - |
| VLM2Vec 7B | 90.7 | 85.8 | 90.8 | 84.7 | - | - |
| UniME 7B | 93.9 | **97.2** | 95.2 | **95.9** | - | - |
| UNITE$_{base}$ 2B | 89.7 | 86.7 | 91.5 | 89.2 | 81.4 | 72.3 |
| UNITE$_{base}$ 7B | 93.3 | 93.2 | **95.5** | 95.6 | **87.2** | **85.8** |

Table 3: Results on the WebVid-CoVR. We take the best variant of baselines in the table.

| Model | WebVid-CoVR-Test (TV→V) | | | |
|---|---|---|---|---|
| | R@1 | R@5 | R@10 | R@50 |
| *Zero-shot Setting* | | | | |
| LanguageBind | 43.2 | 66.3 | 75.2 | - |
| CoVR | 45.5 | 70.5 | 79.5 | 93.3 |
| CoVR-2 | 45.7 | 71.7 | 81.3 | 94.8 |
| TFR-CVR | 51.7 | 75.3 | 80.7 | - |
| *Finetuning Setting* | | | | |
| CoVR | 53.1 | 79.9 | 86.9 | 97.7 |
| CoVR-2 | 59.8 | 83.8 | 91.3 | 98.2 |
| ECDE | 60.1 | 84.3 | 91.3 | 98.7 |
| UNITE$_{instruct}$ 2B | 69.1 | 88.4 | 93.2 | 99.1 |
| UNITE$_{instruct}$ 7B | **72.5** | **90.8** | **95.3** | **99.5** |

Table 4: Results on the MMEB benchmark (Jiang et al., 2024c). We average the scores in each meta-task. We compare numerous recent works, across diverse model scales. "IND" refers to the in-distribution dataset, and "OOD" denotes the out-of-distribution dataset.

| Model | #Parameters | Per Meta-Task Score | | | | Average Score | | |
|---|---|---|---|---|---|---|---|---|
| | | Classification | VQA | Retrieval | Grounding | IND | OOD | Overall |
| #Datasets → | | 10 | 10 | 12 | 4 | 20 | 16 | 36 |
| *Zero-shot Setting* | | | | | | | | |
| Magiclens (ViT-L/14) | 0.4B | 38.8 | 8.3 | 35.4 | 26.0 | 31.0 | 23.7 | 27.8 |
| CLIP (ViT-L/14) | 0.4B | 42.8 | 9.1 | 53.0 | 51.8 | 37.1 | 38.7 | 37.8 |
| OpenCLIP (ViT-L/14) | 0.4B | 47.8 | 10.9 | 52.3 | 53.3 | 39.3 | 40.2 | 39.7 |
| BLIP2 (ViT-L/14) | 1.2B | 27.0 | 4.2 | 33.9 | 47.0 | 25.3 | 25.1 | 25.2 |
| *Finetuning Setting* | | | | | | | | |
| CLIP (ViT-L/14) | 0.4B | 55.2 | 19.7 | 53.2 | 62.2 | 47.6 | 42.8 | 45.4 |
| OpenCLIP (ViT-L/14) | 0.4B | 56.0 | 21.9 | 55.4 | 64.1 | 50.5 | 43.1 | 47.2 |
| E5-V (LLaVA-1.6) | 7B | 39.7 | 10.8 | 39.4 | 60.2 | 34.2 | 33.4 | 33.9 |
| MMRet-MLLM (LLaVA-1.6) | 7B | 56.0 | 57.4 | 69.9 | 83.6 | 68.0 | 59.1 | 64.1 |
| VLM2Vec (Qwen2-VL, high-res) | 7B | 62.6 | 57.8 | 69.9 | 81.7 | 72.2 | 57.8 | 65.8 |
| UniME (LLaVA-1.6) | 7B | 60.6 | 52.9 | 67.9 | 85.1 | 68.4 | 57.9 | 66.6 |
| CAFe (LLaVA-OV) | 7B | 65.2 | **65.6** | 70.0 | **91.2** | **75.8** | 62.4 | 69.8 |
| mmE5 (Llama-3.2-Vision) | 11B | 67.6 | 62.8 | 70.9 | 89.7 | 72.3 | **66.7** | 69.8 |
| IDMR (InternVL2.5) | 26B | 66.3 | 61.9 | 71.1 | 88.6 | 73.4 | 63.9 | 69.2 |
| UNITE$_{instruct}$ 2B | 2B | 63.2 | 55.9 | 65.4 | 75.6 | 65.8 | 60.1 | 63.3 |
| UNITE$_{instruct}$ 7B | 7B | **68.3** | 65.1 | **71.6** | 84.8 | 73.6 | 66.3 | **70.3** |

models with both 2B and 7B parameters. All evaluation datasets are described in Appendix A.2. All experimental results are reported in Recall@1 unless otherwise specified. We refer to Appendix B for more implementation details.

## 4.1 MAIN RESULTS

**Fine-grained Retrieval.** Table 1 demonstrates that our UNITE$_{base}$ achieves state-of-the-art performance and outperforms the existing methods with substantial margins, particularly on CaRe-General and CaRe-Spatial. This excellence stems from the incorporation of fine-grained video-caption pairs during retrieval adaptation, enhancing the feature representation capabilities of LMMs. While our 2B model outperforms all baselines on general and spatial retrieval tasks, its temporal retrieval performance remains moderate. After scaling the model size to 7B, we obtain significant improvements on general, spatial and temporal tasks. Notably, comparing our 2B and 7B models, we observe that model scaling achieves the most substantial relative improvements in temporal retrieval (*e.g.*, 15.7% and 10.8%). It indicates a key insight: *Models with larger sizes are likely to be more advantageous for retrieval tasks related to the temporal aspects of videos.* Moreover, compared with the level that has been achieved in spatial tasks, there is still a great deal of room for improving temporal tasks. In

addition, our UNITE_base achieves leading performance in fine-grained image-text retrieval tasks, as shown in Table 2. It indicates that our carefully designed data composition and allocation strategy indeed enables effective alignment across text, image, and video modalities.

**Instruction-based Retrieval.** As shown in Table 3, UNITE_instruct 2B substantially outperforms existing models on the WebVid-CoVR-Test (Ventura et al., 2024a). Similarly, increasing the scale of model size to 7B can significantly boost the improvement margins. In Table 4, the evaluation on the MMEB benchmark, encompassing 36 datasets across four meta-tasks, demonstrates superior performance of UNITE_instruct against various existing models at different parameter scales. For example, UNITE surpasses both larger-scale models (*e.g.*, mmE5 11B (Chen et al., 2025) and IDMR 26B (Liu et al., 2025)) and models trained with more extensive datasets (*e.g.*, MMRet (Zhou et al., 2024) with 26M image-text retrieval samples). These compelling results across text, image, and video retrieval scenarios can be attributed to our evolving training strategy and MAMCL strategy (See details in Section 4.2).

**Coarse-grained Cross-Modal Retrieval.** We assess our the performance of UNITE on coarse-grained retrieval tasks. As demonstrated in Table 5 and Table 6, our models achieve competitive results on these tasks, including text-image and text-video scenarios. It proves that our data composition strategy effectively balances performance across different modalities on various retrieval tasks.

Table 5: Zero-shot coarse-grained image-text retrieval results on Flickr30K.

| Model | T→I | | I→T | |
|---|---|---|---|---|
| | R@1 | R@5 | R@1 | R@5 |
| OpenCLIP-L | 75.0 | 92.5 | 88.7 | 98.4 |
| MagicLens-L | 79.7 | 95.0 | 89.6 | 98.7 |
| CAFe 7B | 75.3 | 92.6 | 87.5 | 98.2 |
| VLM2Vec 7B | 80.3 | 95.0 | 94.6 | 99.5 |
| LamRA-Ret 7B | 82.8 | - | 92.7 | - |
| UniME 7B | 81.9 | - | 93.4 | - |
| UNITE_base 2B | 80.9 | 95.3 | 89.6 | 98.4 |
| UNITE_base 7B | 86.1 | 96.9 | 94.4 | 99.5 |

Table 6: Zero-shot coarse-grained video-text retrieval results on MSR-VTT, MSVD, and DiDeMo.

| Model | MSR-VTT | | MSVD | | DiDeMo | |
|---|---|---|---|---|---|---|
| | T→V | V→T | T→V | V→T | T→V | V→T |
| InternVideo | 40.7 | 39.6 | 43.4 | 67.6 | 31.5 | 33.5 |
| LanguageBind | 42.1 | 65.9 | 40.1 | 65.4 | 35.6 | 35.6 |
| ViCLIP | 42.4 | 41.3 | 49.1 | 75.1 | 18.4 | 27.9 |
| VLM2Vec 7B | 43.5 | - | 49.5 | - | - | - |
| LamRA 7B | 44.7 | - | 52.4 | - | - | - |
| CaRe 7B | 43.9 | 41.7 | 52.6 | 74.6 | 41.4 | 39.1 |
| UNITE_base 2B | 43.8 | 41.7 | 50.0 | 73.1 | 37.9 | 37.5 |
| UNITE_base 7B | 46.5 | 45.2 | 50.4 | 76.1 | 43.5 | 40.3 |

## 4.2 ABLATION STUDY

**Modal-Aware Masked Contrastive Learning.** We evaluate the effectiveness of MAMCL through comprehensive ablation studies across diverse instruction-based retrieval scenarios. As shown in Table 7 (3→4), we observe that there exists significant performance improvements on MMEB after integrating the MMEB training set. However, the performance degrades on WebVid-CoVR. It verifies our hypothesis that cross-modal interfere might occur among samples with distinct target modalities. Results in Table 7 (1→2, 4→5, 6→7) reveal that MAMCL successfully mitigates these inter-modal effects. Specifically, MAMCL yields substantial improvements in in-distribution (IND) scenarios, validating its effectiveness in scenarios where test samples align with the training distribution. While exhibiting minor fluctuations on out-of-distribution (OOD) datasets, the performance indicates that its generalization capabilities remain well.

## 5 ANALYSIS

In this section, we present a systematic investigation of two practical aspects: (1) the impact of various training data on different retrieval tasks, and (2) efficient training strategies for enhancing the fine-grained retrieval capabilities of LMMs.

**Training Data Composition** Understanding the optimal composition of training data for text-image-video retrieval scenarios remains an open research problem that warrants systematic investigation.

Table 7: Ablation study of our proposed MAMCL. We show results of various settings on the MMEB and WebVid-CoVR test sets. Gray indicates without fine-tuning on the corresponding training set. **Avg** refers to the average of the overall score on MMEB and the R@1 score on WebVid-CoVR.

| ID | Setting | | | MMEB | | | WebVid-CoVR | | | Avg |
|---|---|---|---|---|---|---|---|---|---|---|
| | MMEB | CoVR | MAMCL | IND | OOD | Overall | R@1 | R@5 | R@10 | |
| *2B parameters* | | | | | | | | | | |
| 1 | ✓ | | | 64.4 | 60.1 | 62.5 | 49.6 | 74.5 | 82.1 | - |
| 2 | ✓ | | ✓ | 65.5 (+1.1) | 60.2 (+0.1) | 63.1 (+0.6) | 48.0 | 73.4 | 80.9 | - |
| 3 | | ✓ | | 36.1 | 34.9 | 35.6 | 69.2 | 89.4 | 93.6 | - |
| 4 | ✓ | ✓ | | 64.8 | 60.3 | 62.8 | 67.4 | 88.0 | 92.6 | 65.1 |
| 5 | ✓ | ✓ | ✓ | 65.8 (+1.0) | 60.1 (-0.2) | 63.3 (+0.5) | 69.1 (+1.7) | 88.4 (+0.4) | 93.2 (+0.6) | 66.2 (+1.1) |
| *7B parameters* | | | | | | | | | | |
| 6 | ✓ | ✓ | | 73.3 | 65.8 | 70.0 | 71.4 | 91.1 | 94.7 | 70.7 |
| 7 | ✓ | ✓ | ✓ | 73.6 (+0.3) | 66.3 (+0.5) | 70.3 (+0.3) | 72.5 (+1.1) | 90.8 (-0.3) | 95.3 (+0.6) | 71.4 (+0.7) |

We conduct comprehensive experiments utilizing Text-Text (TT), Text-Image (TI), and Text-Video (TV) datasets, both independently and in various combined scenarios (See details in Appendix B.3). Finally, the evaluation covers coarse-grained, fine-grained, and instruction-based retrieval tasks.

*Video-text pairs prove to be superior training data for general cross-modal retrieval.* As shown in the "Coarse" and "Fine" results in Table 8 (See more details in Table 18), across both coarse-grained and fine-grained tasks, the TV-only training pattern consistently achieves the best performance. Notably, in image-text retrieval tasks, training solely with TV data outperforms training that uses only TI data, which contradicts intuitive expectations in image-text retrieval. Compared with static images, videos provide richer and situational dynamic visual context, including event progression, and temporal evolution, enabling broader and more generalizable vision–language alignment. Moreover, an image can be regarded as a special case of a video (*i.e.*, a single static frame) (Bain et al., 2021). Once the model has mastered alignment for complex video–text semantics, transferring to simpler static visual signals becomes straightforward, leading to superior performance even on image–text retrieval. This finding highlight the need to reconsider traditional data selection strategies and encourages further exploration into how different data types interact with training in cross-modal retrieval.

Table 8: Results of various training data composition on different retrieval tasks in retrieval adaptation stage. To ensure fairness, the total data size for all configurations is 600K. All scores of cross-modal retrieval are the average of R@1 in zero-shot setting. The tested datasets include (1) coarse-grained image-text datasets (Flickr30K, MSCOCO), video-text datasets (MSR-VTT, MSVD); (2) fine-grained image-text dataset (DOCCI), video-text dataset (CaReBench); and (3) instruction-based datasets (MMEB, WebVid-CoVR).

| Setting | | | Coarse I-T | | Coarse V-T | | Fine I-T | | Fine V-T | | MMEB | | | CoVR |
|---|---|---|---|---|---|---|---|---|---|---|---|---|---|---|
| TT | TI | TV | T→I | I→T | T→V | V→T | T→I | I→T | T→V | V→T | IND | OOD | Overall | R@1 |
| ✓ | | | 53.5 | 64.1 | 37.8 | 45.8 | 69.1 | 65.4 | 45.5 | 52.4 | 61.9 | 58.5 | 60.4 | 64.4 |
| | ✓ | | 55.4 | 68.9 | 40.5 | 51.0 | 75.8 | 71.8 | 57.9 | 62.9 | 62.6 | 58.3 | 60.7 | **66.5** |
| | | ✓ | **60.2** | **73.8** | **44.3** | **56.0** | **79.8** | **74.9** | **65.8** | **68.7** | 62.6 | 59.1 | 61.1 | 65.6 |
| ✓ | ✓ | | 55.6 | 67.6 | 41.6 | 50.8 | 74.8 | 70.3 | 56.8 | 58.8 | **63.8** | **59.9** | **62.1** | 65.4 |
| ✓ | | ✓ | 56.9 | 65.1 | 43.4 | 54.6 | 76.3 | 70.1 | 62.3 | 64.1 | 62.1 | 57.7 | 60.2 | 65.6 |
| | ✓ | ✓ | 58.3 | 71.8 | 43.7 | 55.9 | 77.8 | 73.0 | 65.7 | 68.4 | 63.0 | 59.2 | 61.3 | 65.8 |
| ✓ | ✓ | ✓ | 58.1 | 67.8 | 43.0 | 54.6 | 76.5 | 70.9 | 61.3 | 61.2 | 62.7 | 59.0 | 61.0 | 64.8 |

*Text-text and text-image pairs are essential for instruction-following tasks.* To further investigate the impact of data composition on broader retrieval tasks, we conducted comprehensive experiments on instrcution-based retrieval tasks. TT+TI training overall outperforms other combinations on instruction-based retrieval tasks, including TV-only configuration that excel in general cross-modal retrieval tasks, as shown in the "MMEB" and "CoVR" results in Table 8 (See more details in Table 19). This observation can be attributed to two key factors: (1) Text-text pairs enhance linguistic understanding and logical reasoning capabilities, establishing a solid and comprehensive foundation for interpreting complex retrieval instructions. (2) Text-image pairs provide precise multimodal

alignment information, empowering more focused semantic connections compared to video content. These factors enable the model to capture detailed vision-language correspondences that are crucial for instruction following.

**Effectively Utilizing Fine-Grained Video-Caption Data** Recent advances in video LMMs have produced powerful captioning models (*e.g.*, Tarsier (Wang et al., 2024a), AuroraCap (Chai et al., 2024)) and fine-grained datasets like LLaVA-Video-178K (Zhang et al., 2024f). While CaRe has shown that fine-tuning LMMs with these video-caption pairs prior to retrieval adaptation can significantly boost the performance of fine-grained video retrieval, a notable limitation exists in its retrieval adaptation phase, which depends solely on text-text pairs. This raises two questions: (1) Does the *fine-grained alignment* continue to yield effective results when video-text pairs are integrated into the retrieval adaptation process? and (2) How can we optimally utilize fine-grained video-text pairs to achieve the greatest possible enhancements in performance?

Table 9: Ablation study on fine-grained alignment stage (**Align**) across distinct retrieval-adaptation data (**Retrieval**). The reported scores are Recall@1 of zero-shot results. The composition and volume of retrieval-adaptation data used for ID 1-6 are consistent with those in Table 8 in Appendix C.3.

| ID | Setting | | Coarse-grained Video-Text Retrieval | | | | Fine-grained Video-Text Retrieval | | | | | | |
|---|---|---|---|---|---|---|---|---|---|---|---|---|---|
| | | | MSR-VTT | | MSVD | | CaRe-General | | CaRe-Spatial | | CaRe-Temporal | | Avg |
| | Align | Retrieval | T→V | V→T | T→V | V→T | T→V | V→T | T→V | V→T | T→V | V→T | |
| 1 | ✗ | TT | 32.9 | 31.5 | 42.7 | 60.1 | 45.5 | 52.4 | 47.7 | 51.3 | 33.1 | 37.7 | 43.5 |
| 2 | ✓ | TT | 33.5 (+0.6) | 30.8 (-0.7) | 42.5 (-0.2) | 60.1 (-) | 50.1 (+4.6) | 51.5 (-0.9) | 51.3 (+3.6) | 51.4 (+0.1) | 34.8 (+1.7) | 37.7 (-) | 44.4 (+0.9) |
| 3 | ✗ | TV | **41.5** | **41.7** | 47.0 | **70.3** | 65.8 | 68.7 | 68.3 | 67.3 | 42.0 | **47.1** | 56.0 |
| 4 | ✓ | TV | 39.6 (-1.9) | 40.4 (-1.3) | **47.4** (+0.4) | 70.3 (-) | 68.5 (+2.7) | 68.1 (-0.6) | 67.6 (-0.7) | 65.9 (-1.4) | 41.6 (-0.4) | 45.3 (-1.8) | 55.5 (-0.5) |
| 5 | ✗ | TT+TI+TV | 40.0 | 39.2 | 46.0 | 69.9 | 61.3 | 61.2 | 63.0 | 59.6 | 40.5 | 42.7 | 52.3 |
| 6 | ✓ | TT+TI+TV | 38.1 (-1.9) | 39.0 (-0.2) | 45.6 (-0.4) | 69.1 (-0.8) | 59.1 (-2.2) | 60.2 (-1.0) | 62.1 (-0.9) | 58.8 (-0.8) | 39.7 (-0.8) | 43.1 (+0.4) | 51.5 (-0.8) |
| 7 | ✗ | Fine TV | 34.8 | 35.6 | 42.3 | 69.6 | **81.2** | **79.9** | **81.0** | **80.5** | **42.9** | 46.0 | **59.4** |
| 8 | ✗ | TV + Fine TV | 39.9 | 39.1 | 40.6 | 69.6 | 79.2 | 79.0 | 79.4 | 76.7 | 41.8 | 45.7 | **59.4** |

To solve these two questions, we conduct extensive experiments across diverse settings (See details in Appendix B.4). Our research endeavors focus on evaluating the effectiveness of *fine-grained alignment*. This is achieved by performing next token prediction fine-tuning on 500K fine-grained video-caption instances sourced from Tarsier2-Recap-585K (Yuan et al., 2025) with various configurations. Initial experiments validate the findings in CaRe (Xu et al., 2024b): when text-text (TT) data is employed for retrieval adaptation, fine-grained alignment significantly boosts performance, especially in fine-grained video-text retrieval tasks (See Table 9, 1→2). However, when text-video (TV) pairs are involved in retrieval adaptation, fine-grained alignment surprisingly leads to performance degradation (See Table 9, 3→4, 5→6). Our results reveal several key insights:

- During the retrieval adaptation process, leveraging TV pairs yields far more significant performance improvements than those obtained through fine-grained alignment.
- The exclusive employment of fine-grained video-text pairs during the retrieval adaptation process leads to a substantial enhancement in CaReBench performance. However, it causes a severe degradation in the model's coarse-grained retrieval capabilities (See Table 9, 7).
- By incorporating fine-grained TV pairs into general TV data (See Table 9, 8), we can achieves a balanced performance. It enables the model to obtain competitive results in both coarse- and fine-grained video-text retrieval tasks.

Thus, these observations lead to an insight: *During the retrieval adaptation, the direct incorporation of fine-grained video-caption pairs has been shown to be far more effective than the implementing a isolated fine-grained alignment stage*.

## 6 CONCLUSION

In this work, we introduce UNITE, a universal multimodal embedding framework that enables the seamlessly integration of text, image, and video modalities. Through systematic analysis of how training with varied data compositions affects the final retrieval performance, we observe novel insights that have not received limited exploration in image-text and video-text retrieval scenarios. Based on these insights, we propose an data composition and allocation strategy, and introduce MAMCL to mitigate inter-instance competition while maintaining balance the representation learning

across text, images, and videos. Extensive experimental results demonstrate that UNITE achieves state-of-the-art results on 40+ tasks spanning coarse-grained, fine-grained, and instruction-based retrieval scenarios. We believe that our work advances the development of unified multimodal retrieval and provides valuable insights for future research.

ETHICS STATEMENT

Our research focuses on developing methods for multimodal information retrieval using publicly available datasets (e.g., MSMARCO, LAION, MMEB, WebVid-CoVR). We have not collected any human subjects, private information, or sensitive data. We take care to use datasets that are widely adopted in the community and ensured compliance with their licenses. Our methods aim to advance retrieval performance and do not pose foreseeable risks of misuse or societal harm beyond those already inherent to large multimodal models.

REPRODUCIBILITY STATEMENT

We have made efforts to ensure the reproducibility of our work. All model architectures, training strategies, and evaluation protocols are described in detail in Section 4 and 5 of the main text, with dataset compositions, and implementation details provided in Appendix A and B. Comprehensive ablation studies and analyses (Section 4 and 5) further validate the robustness of our findings. Notably, all evaluations are free from randomness, as we have fixed the random seed and employed no random sampling schemes or temperature schedules during inference. To facilitate reproducibility, we provide an anonymous code repository and scripts for training and evaluation.

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

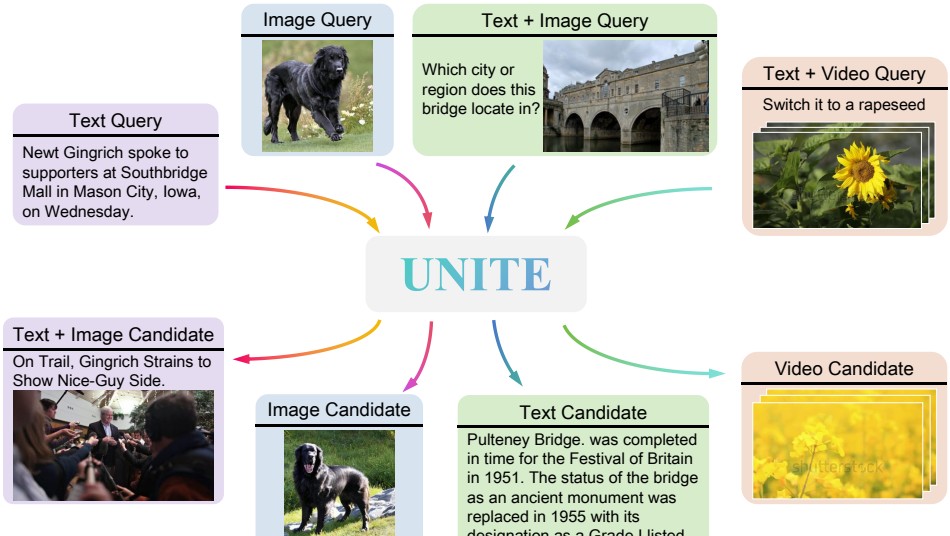

Figure 3: We develop a universal multimodal embedder UNITE, allowing for a unified representation of arbitrary multimodal contents.

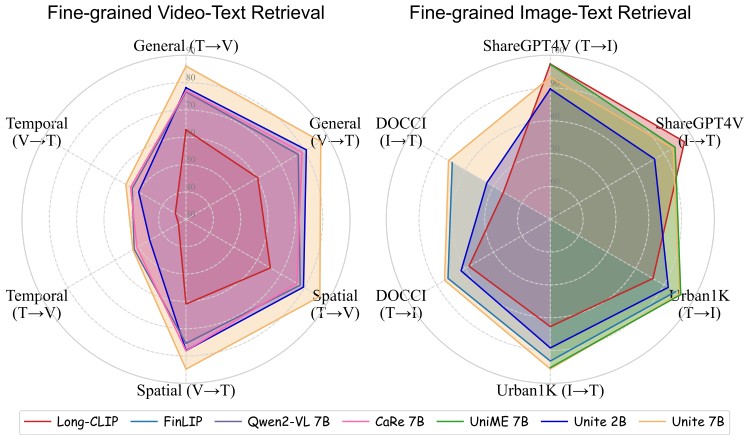

Figure 4: Performance comparison on fine-grained video-text benchmark (CaReBench (Xu et al., 2024b)) and image-text benchmarks (ShareGPT4V (Chen et al., 2024a), Urban1K (Zhang et al., 2024a), DOCCI (Onoe et al., 2024)). Our UNITE achieves the overall optimal performance.

## A DATA

### A.1 TRAINING DATA COMPOSITION

Our thoroughly analysis across coarse-grained, fine-grained, and instruction-based retrieval tasks reveals that text-video pairs are particularly effective for cross-modal retrieval, while text-text and text-image pairs are essential for instruction-based retrieval (Section 5). Based on these findings, we carefully curate the retrieval adaptation data to achieve a balanced performance across different modalities. The detailed data composition is illustrated in Figure 5.

For instruction tuning, surpassing existing works that solely rely on supervised learning on MMEB (Jiang et al., 2024c), we combine 20 instruction-based image-text retrieval datasets from MMEB with a subset of WebVid-CoVR (Ventura et al., 2024a) to optimize performance across text, image, and video modalities. The detailed data composition is illustrated in Figure 6.

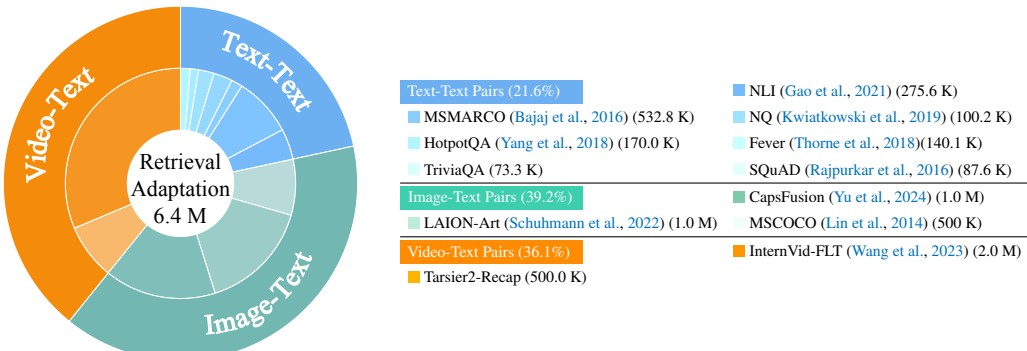

Figure 5: **Retrieval Adaptation 6.4M.** Left: Data Distribution within Each Category. The outer circle shows the distribution of all data categories and the inner circle shows the distribution of data subsets. Right: The detailed quantities of datasets.

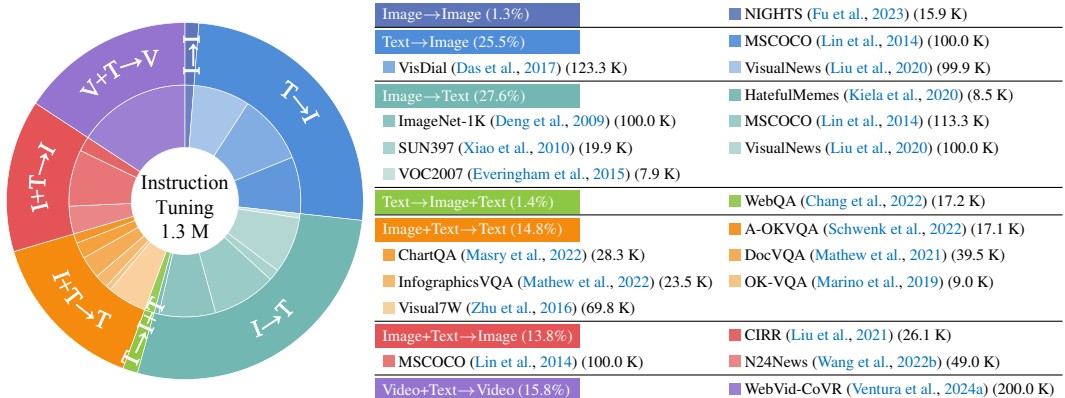

Figure 6: **Instruction Tuning 1.3M.** Left: Data Distribution within Each Category. The outer circle shows the distribution of all data categories and the inner circle shows the distribution of data subsets. Right: The detailed quantities of datasets.

## A.2 EVALUATION DATASETS

We provide brief descriptions of all evaluation benchmarks, and their statistics are shown in Table 10.

Table 10: Evaluation benchmark statistics. MMEB is a comprehensive benchmark and its statistics are shown in Table 11. #Text/Instruction and #Image/Video denote the test text/instruction count and image/video pool size, respectively.

| Benchmark | Query→Target | Zero-shot | #Text/Instrcution | #Image/Video |
|---|---|---|---|---|
| *Coarse-grained Retrieval* | | | | |
| Flickr30K (Plummer et al., 2015) | T→I, I→T | ✓ | 1,000 | 5,000 |
| MSR-VTT (Xu et al., 2016) | T→V, V→T | ✓ | 1,000 | 1,000 |
| MSVD (Chen & Dolan, 2011) | T→V, V→T | ✓ | 670 | 27,763 |
| DiDeMo (Anne Hendricks et al., 2017) | T→V, V→T | ✓ | 1,004 | 1,004 |
| *Fine-grained Retrieval* | | | | |
| ShareGPT4V (Plummer et al., 2015) | T→I, I→T | ✓ | 1,000 | 1,000 |
| Urban1K (Zhang et al., 2024a) | T→I, I→T | ✓ | 1,000 | 1,000 |
| DOCCI (Onoe et al., 2024) | T→I, I→T | ✓ | 5,000 | 5,000 |
| CaRe (Xu et al., 2024b) | T→V, V→T | ✓ | 1,000 | 1,000 |
| *Instruction-based Retrieval* | | | | |
| MMEB (Plummer et al., 2015) | 36 tasks | ✗ | - | - |
| WebVid-CoVR (Zhang et al., 2024a) | T+V→V | ✗ | 2,556 | 2,556 |

### A.2.1 Fine-grained Retrieval Datasets

**CaReBench** (Xu et al., 2024b) consists of 1,000 video-caption pairs with hierarchical annotations covering overall summary, static objects, dynamic actions, and miscellaneous aspects. Its distinctive feature lies in the manually annotated spatial and temporal information. We evaluate fine-grained video-text retrieval through three tasks: General, Spatial, and Temporal.

**ShareGPT4V** (Chen et al., 2024a) features comprehensive image-text pairs with rich descriptions of world knowledge, object properties, spatial relationships, and aesthetic elements. The dataset comprises 100K GPT4-Vision generated captions and 1.2M model-expanded captions, averaging 942 characters in length. Following Long-CLIP (Zhang et al., 2024a), we utilize 1K instances for testing.

**Urban1K** (Zhang et al., 2024a) is derived from Urban-200 and expanded to 1,000 urban scene image-text pairs. Each image is paired with a GPT-4V generated caption (averaging 101 words) detailing object types, colors, and spatial relationships. The dataset challenges models with visually similar urban scenes, requiring fine-grained cross-modal understanding.

**DOCCI** (Onoe et al., 2024) (Descriptions of Connected and Contrasting Images) contains 15,000 images with human-annotated descriptions (averaging 136 words). Curated by a single researcher, it evaluates spatial relations, counting, text rendering, and world knowledge comprehension. The dataset features contrast sets with subtle variations in object arrangements. We evaluate using the official 5K test split.

### A.2.2 Instruction-based Retrieval Datasets

**MMEB** (Jiang et al., 2024c) is a comprehensive multimodal embedding benchmark comprising 36 datasets across four meta-tasks: classification, visual question answering, retrieval, and visual grounding. The benchmark is strategically divided into 20 in-distribution training datasets and 16 out-of-distribution evaluation datasets. All tasks are formulated as ranking problems with 1,000 candidates, where models process instruction-guided queries (text, images, or both) to select correct targets.

The meta-tasks are structured as follows:

- **Classification**: Queries combine instructions with images (and optional text) to predict class labels.
- **Visual Question Answering**: Queries include instructions, images, and questions, with answers as targets.
- **Information Retrieval**: Both queries and targets can be multimodal combinations.
- **Visual Grounding**: Queries pair instructions with full images to locate specific objects, using cropped regions as candidates.

MMEB spans diverse domains (common, news, Wikipedia, web, fashion) and supports various instruction types from object recognition to retrieval tasks. Performance is evaluated using Precision@1, measuring the accuracy of selecting the correct candidate from 1,000 options. The benchmark's comprehensive design makes it an ideal testbed for universal multimodal embeddings.

**WebVid-CoVR-Test** (Ventura et al., 2024a) is a manually verified benchmark for Composed Video Retrieval (CoVR), containing 2,435 high-quality video-text-video triplets. Each triplet consists of a query video, a modification text describing desired changes, and a target video. The benchmark is derived from WebVid10M, specifically excluding WebVid2M content to ensure evaluation integrity. The dataset features diverse modification texts (averaging 4.8 words) and videos (averaging 16.8 seconds), covering a wide range of content variations.

### A.2.3 Coarse-grained Retrieval Datasets

**Flickr30K** (Plummer et al., 2015) consists of 31K images collected from Flickr, with each image paired with five human-annotated captions describing its content. The dataset covers diverse everyday scenarios and human activities in natural settings. Following common practice (Karpathy & Fei-Fei, 2015), we use the standard split with 1,000 images for testing.

**MSRVTT** (Xu et al., 2016) comprises 10K video clips paired with 200K captions, covering diverse topics including human activities, sports, and natural landscapes. For text-to-video retrieval evaluation, we adopt the standard `1K-A split` following prior works.

Table 11: The statistics of MMEB: 36 datasets across 4 meta-task categories, with 20 in-distribution datasets used for training and 16 out-of-distribution datasets used exclusively for evaluation.

| Meta-Task | Dataset | Query→Target | Distribution Type | #Training | #Eval | #Candidates |
|---|---|---|---|---|---|---|
| Classification (10 Tasks) | ImageNet-1K | I→T | IND | 100K | 1000 | 1000 |
| | N24News | I + T→I | IND | 49K | 1000 | 24 |
| | HatefulMemes | I→T | IND | 8K | 1000 | 2 |
| | VOC2007 | I→T | IND | 8K | 1000 | 20 |
| | SUN397 | I→T | IND | 20K | 1000 | 397 |
| | Place365 | I→T | OOD | - | 1000 | 365 |
| | ImageNet-A | I→T | OOD | - | 1000 | 1000 |
| | ImageNet-R | I→T | OOD | - | 1000 | 200 |
| | ObjectNet | I→T | OOD | - | 1000 | 313 |
| | Country-211 | I→T | OOD | - | 1000 | 211 |
| VQA (10 Tasks) | OK-VQA | I + T→T | IND | 9K | 1000 | 1000 |
| | A-OKVQA | I + T→T | IND | 17K | 1000 | 1000 |
| | DocVQA | I + T→T | IND | 40K | 1000 | 1000 |
| | InfographicVQA | I + T→T | IND | 24K | 1000 | 1000 |
| | ChartQA | I + T→T | IND | 28K | 1000 | 1000 |
| | Visual7W | I + T→T | IND | 70K | 1000 | 1000 |
| | ScienceQA | I + T→T | OOD | - | 1000 | 1000 |
| | VizWiz | I + T→T | OOD | - | 1000 | 1000 |
| | GQA | I + T→T | OOD | - | 1000 | 1000 |
| | TextVQA | I + T→T | OOD | - | 1000 | 1000 |
| Retrieval (12 Tasks) | VisDial | T→I | IND | 123K | 1000 | 1000 |
| | CIRR | I + T→I | IND | 26K | 1000 | 1000 |
| | VisualNews_t2i | T→I | IND | 100K | 1000 | 1000 |
| | VisualNews_i2t | I→T | IND | 100K | 1000 | 1000 |
| | MSCOCO_t2i | T→I | IND | 100K | 1000 | 1000 |
| | MSCOCO_i2t | I→T | IND | 113K | 1000 | 1000 |
| | NIGHTS | I→I | IND | 16K | 1000 | 1000 |
| | WebQA | T→I + T | IND | 17K | 1000 | 1000 |
| | OVEN | I + T→I + T | OOD | - | 1000 | 1000 |
| | FashionIQ | I + T→I | OOD | - | 1000 | 1000 |
| | EDIS | T→I + T | OOD | - | 1000 | 1000 |
| | Wiki-SS-NQ | T →I | OOD | - | 1000 | 1000 |
| Visual Grounding (4 Tasks) | MSCOCO | I + T→I | IND | 100K | 1000 | 1000 |
| | Visual7W-Pointing | I + T→I | OOD | - | 1000 | 1000 |
| | RefCOCO | I + T→I | OOD | - | 1000 | 1000 |
| | RefCOCO-Matching | I + T→I + T | OOD | - | 1000 | 1000 |

**MSVD** (Chen & Dolan, 2011) features 1,970 videos, with approximately 40 captions annotated per video.

**DiDeMo** (Anne Hendricks et al., 2017) features 10K long-form Flickr videos, where each video is annotated with four temporally ordered sentences. Following previous works, we concatenate these sentences for paragraph-to-video retrieval evaluation using the official split.

## B  MORE IMPLEMENTATION DETAILS

We simply incorporate LoRA (Hu et al., 2022) module into LMM, with a rank of 8, and gradually unlock its retrieval capabilities. We employ a temperature of 0.03 for contrastive learning, with learning rates of 1e-4 and 2e-5 for retrieval adaptation and instruction tuning phases, respectively. Following Qwen2-VL (Wang et al., 2024b), we process images with dynamic resolution, constraining the number of image tokens to (256, 1,280). For video processing, we sample frames at 1fps with a 12-frame cap, limiting each frame's maximum number of tokens to 98. Our experiments are conducted on 64 NVIDIA A100 GPUs for training and 8 NVIDIA H100 GPUs for evaluation. The following experiments adopt these default settings unless specified otherwise.

## B.1 STAGE1: RETRIEVAL ADAPTATION

During retrieval adaptation, we follow the standard InfoNCE loss (*i.e.*, Eq. (1)) and employ a bidirectional contrastive learning strategy to maximize the utilization of training data. Taking image-text pairs as an example, we simultaneously optimize both text-to-image and image-to-text retrieval directions. The final loss function is computed as the average of these bidirectional retrieval processes:

$$\mathcal{L}_{\text{bi}} = \frac{1}{2} \left[ -\frac{1}{N} \sum_{n=1}^{N} \log \frac{\exp\left(\cos(\mathbf{q}_n, \mathbf{c}_n^+)/\tau\right)}{\sum_{j=1}^{N} \exp\left(\cos\left(\mathbf{q}_n, \mathbf{c}_j\right)/\tau\right)} - \frac{1}{N} \sum_{n=1}^{N} \log \frac{\exp\left(\cos(\mathbf{c}_n^+, \mathbf{q}_n)/\tau\right)}{\sum_{j=1}^{N} \exp\left(\cos\left(\mathbf{c}_n^+, \mathbf{q}_j\right)/\tau\right)} \right]$$

The training data distribution and hyperparameters are shown in Figure 5 and Table 12, respectively.

## B.2 STAGE2: INSTRUCTION TUNING

During instruction tuning, we observe that jointly training on instruction-based retrieval data from different modalities (image-text and video-text) may lead to performance fluctuations, with potential degradation in certain domains. To address this challenge and maintain consistent performance across modalities, we propose MAMCL (Modality-Aware Masked Contrastive Learning) to balance the learning dynamics between different modal data. This approach helps optimize the overall multimodal retrieval capabilities while preserving domain-specific performance. The detailed data distribution and training parameters are presented in Figure 6 and Table 13.

Table 12: Training hyperparameters and computational requirements for UNITE_base.

| Hyperparameter | UNITE 2B | UNITE 7B |
|---|---|---|
| *Stage1: Retrieval Adaptation for UNITE_base* | | |
| Training Samples | 6.4M | |
| Batch Size | 4,096 | 1,024 |
| Learning rate | $1 \times 10^{-4}$ | |
| Optimizer | AdamW | |
| Learning Rate Decay | cosine | |
| Warmup Ratio | 0.03 | |
| LoRA Rank | 8 | |
| LoRA Alpha | 16 | |
| Temperature $\tau$ | 0.03 | |
| Epochs | 1 | |
| GPU Configuration | 64×A100 | |
| Training Time | 7 hours | 21 hours |

Table 13: Training hyperparameters and computational requirements for UNITE_instruct.

| Hyperparameter | UNITE 2B | UNITE 7B |
|---|---|---|
| *Stage2: Instruction Tuning for UNITE_instruct* | | |
| Training Samples | 1.3M | |
| Batch Size | 4,096 | 1,024 |
| Learning rate | $2 \times 10^{-5}$ | |
| Optimizer | AdamW | |
| Learning Rate Decay | cosine | |
| Warmup Ratio | 0.03 | |
| LoRA Rank | 8 | |
| LoRA Alpha | 64 | |
| Temperature $\tau$ | 0.03 | |
| Epochs | 1 | |
| GPU Configuration | 64×A100 | |
| Training Time | 2 hours | 6 hours |

## B.3 ANALYSIS: TRAINING DATA COMPOSITION

We conduct comprehensive experiments utilizing Text-Text (TT), Text-Image (TI), and Text-Video (TV) datasets, both independently and in various combined scenarios. Specifically, we collect TT data from MSMARCO (Bajaj et al., 2016) and NLI (Gao et al., 2021), TI data from CapsFusion (Yu et al., 2024), and TV data from InternVid (Wang et al., 2023). To ensure fair comparisons, we maintain a consistent total dataset size of 600K instances across all configurations, with mixed-pattern settings employing equal distribution of different constituent types (*e.g.*, 200K per dataset in TT+TI+TV).

The experimental pipeline consists of two phases. In the first phase, we finetune Qwen2-VL-2B (Wang et al., 2024b) using 600K instances across seven distinct configurations, following the hyperparameters specified in Table 12. The second phase involves instruction tuning, where we independently finetune the retrieval-adapted model using the complete MMEB (Jiang et al., 2024c) training set and 500K instances from WebVid-CoVR (Ventura et al., 2024a), respectively, adhering to the hyperparameters outlined in Table 13.

To thoroughly investigate the impact of various data compositions on retrieval performance, we evaluate across three distinct retrieval scenarios, including (1) coarse-grained image-text datasets (Flickr30K (Plummer et al., 2015), MSCOCO (Lin et al., 2014)), video-text datasets (MSR-VTT (Xu et al., 2016), MSVD (Chen & Dolan, 2011)); (2) fine-grained image-text dataset (DOCCI (Onoe

et al., 2024)), video-text dataset (CaReBench (Xu et al., 2024b)); and (3) instruction-based datasets (MMEB (Jiang et al., 2024c), WebVid-CoVR (Ventura et al., 2024a)). The raw results are shown in Table 18 and Table 19.

### B.4 Analysis: Effectively Utilizing Fine-Grained Video-Caption Data

To investigate efficient strategies for fine-grained video-caption training, we conduct preliminary experiments using Tarsier2-Recap-585K (Yuan et al., 2025). Extending the exploration scope of CaRe's (Xu et al., 2024b) work, we examine the effectiveness of fine-grained alignment across broader data compositions (TT, TV, and TT+TI+TV). This expanded investigation aims to understand the generalizability of *fine-grained alignment* (*i.e.*, fine-tuning LMM through next token prediction using fine-grained video-caption pairs) in diverse multimodal scenarios.

The experimental configurations in Table 9 (IDs 1-6) utilize 500K instances from Tarsier2-Recap for fine-grained alignment, maintaining consistent retrieval adaptation data as described in Section B.3. Configuration ID 7 omits the fine-grained alignment, instead directing all 500K fine-grained video-caption pairs to retrieval adaptation. Similarly, ID 8 bypasses fine-grained alignment and employs a balanced training set with 600K instances, comprising 300K instances each from InternVid and Tarsier2-Recap.

For fine-grained alignment, we leverage `ms-swift`[1] to train LoRA modules of Qwen2-VL-2B, configured with a rank of 16 and an alpha of 32. We set the batch size to 128 and the learning rate to 1e-4. For video items, we sample frames at 1fps with a 16-frame cap, limiting each frame's maximum number of tokens to 192.

Notably, unified contrastive learning demonstrates superiority not only in performance but also in computational efficiency. Our empirical analysis reveals significant training speed advantages: processing 500K instances from Tarsier2-Recap requires 4.2 hours for fine-grained alignment and merely 0.9 hours for retrieval adaptation, highlighting the method's computational efficiency while maintaining superior performance.

## C More Experiments

### C.1 Baselines

We compare our framework against leading models specialized for each benchmark. Notably, these task-specific models often excel only on their designated tasks but perform poorly or lack support on others. In contrast, our unified framework consistently surpasses these specialized baselines across diverse tasks. Below we list the baselines adopted for each task:

- **Coarse-grained Video-Text Retrieval.** InternVideo (Wang et al., 2022a), LanguageBind (Zhu et al., 2023a), ViCLIP (Wang et al., 2023), VLM2Vec (Jiang et al., 2024c), LamRA (Liu et al., 2024d), CaRe (Xu et al., 2024b).
- **Coarse-grained Image-Text Retrieval.** OpenCLIP (Cherti et al., 2023), MagicLens (Zhang et al., 2024c), CAFe (Yu et al., 2025), VLM2Vec (Jiang et al., 2024c), LamRA (Liu et al., 2024d), UniME (Gu et al., 2025).
- **Fine-grained Video-Text Retrieval.** CLIP (Radford et al., 2021), LanguageBind (Zhu et al., 2023a), Long-CLIP (Zhang et al., 2024a), InternVideo2 (Wang et al., 2024c), LLaVA-Next-Video (Zhang et al., 2024e), MiniCPM-V 2.6 (Yao et al., 2024), InternVL2 (Chen et al., 2024b), Tarsier[†] (Wang et al., 2024a), Qwen2-VL (Wang et al., 2024b), CaRe (Xu et al., 2024b).
- **Fine-grained Image-Text Retrieval.** CLIP (Radford et al., 2021), OpenCLIP L/14 (Cherti et al., 2023), Long-CLIP (Zhang et al., 2024a), EVA-CLIP (Sun et al., 2023), FineLIP (Asokan et al., 2025), MATE (Jang et al., 2024), E5-V (Jiang et al., 2024b), VLM2Vec (Jiang et al., 2024c), UniME (Gu et al., 2025).
- **Instruction-based Video-Text Retrieval.** LanguageBind (Zhu et al., 2023a), CoVR (Ventura et al., 2024a), CoVR-2 (Ventura et al., 2024b), TFR-CVR (Hummel et al., 2024), ECDE (Thawakar et al., 2024)

---

[1] https://github.com/modelscope/ms-swift

- **Instruction-based Image-Text Retrieval.** CLIP (Radford et al., 2021), OpenCLIP (Cherti et al., 2023), E5-V (Jiang et al., 2024b), MMRet-MLLM (Zhou et al., 2024), VLM2Vec (Jiang et al., 2024c), UniME (Gu et al., 2025), CAFe (Yu et al., 2025), mmE5 (Chen et al., 2025), IDMR (Liu et al., 2025).

## C.2 Detailed Main Results

Detailed experimental results on CaReBench, MMEB, and coarse-grained video-text retrieval benchmarks are provided in Tables 15, 16, and 17, in correspondence with Tables 1, 4, and 6. Table 20 provides the results of MMEB specifically for the 36 tasks. Figure 4 shows the visualization of the fine-grained retrieval results.

## C.3 Detailed Results for Analysis

Tables 18, 19 show the raw results of Table 8.

## C.4 Computational Efficiency of MAMCL

In this section, we provide a comprehensive analysis of the computational efficiency of the proposed MAMCL strategy, including theoretical complexity analysis and empirical runtime measurements.

**Theoretical Complexity.** Let $N$ denote the batch size, $K$ denote the number of negative candidates, and $D$ denote the embedding dimension (for UNITE 2B, $D = 1536$; for UNITE 7B, $D = 3584$). For standard InfoNCE, the time complexity of computing the similarity matrix is: $O(N \cdot (1 + K) \cdot D)$. MAMCL introduces a modality-aware mask with computation cost: $O(N \cdot (1 + K))$, followed by the same similarity matrix computation as InfoNCE: $O(N \cdot (1 + K))$. Since $D \gg 1$, the additional overhead of applying the mask is negligible compared to the dominant similarity computation term. Therefore, the overhead of MAMCL is negligible compared to the dominant similarity matrix computation cost.

**Actual Runtime Analysis.** To validate our theoretical analysis, we measure the actual training time with and without MAMCL on the instruction tuning stage. The experimental setup follows the configuration described in Table 13. Table 14 presents the wall-clock training time for 2B and 7B sizes. The results demonstrate that MAMCL introduces negligible overhead.

Table 14: Training time comparison with and without MAMCL on the instruction tuning stage.

|  | UNITE 2B | UNITE 7B |
|---|---|---|
| w/o MAMCL | 2:15:33 | 6:17:29 |
| w/ MAMCL | 2:15:10 | 6:19:54 |
| Difference | -23s (-0.3%) | +145s (+0.6%) |

Table 15: Detailed results of zero-shot performance on CaReBench (Xu et al., 2024b).

| Model | CaRe-General | | | | | | CaRe-Spatial | | | | | | CaRe-Temporal | | | | | |
|---|---|---|---|---|---|---|---|---|---|---|---|---|---|---|---|---|---|---|
| | T→V | | | V→T | | | T→V | | | V→T | | | T→V | | | V→T | | |
| | R@1 | R@5 | R@10 | R@1 | R@5 | R@10 | R@1 | R@5 | R@10 | R@1 | R@5 | R@10 | R@1 | R@5 | R@10 | R@1 | R@5 | R@10 |
| *CLIP-based Models* | | | | | | | | | | | | | | | | | | |
| CLIP L/14 | 51.2 | 83.4 | 90.6 | 54.7 | 86.9 | 93.6 | 49.0 | 81.9 | 91.4 | 55.4 | 85.6 | 93.0 | 33.5 | 70.3 | 84.0 | 39.7 | 76.2 | 87.9 |
| LanguageBind | 64.3 | 91.0 | 96.3 | 59.5 | 88.0 | 95.0 | 64.7 | 90.8 | 96.8 | 61.0 | 87.2 | 94.5 | 39.8 | 77.3 | 90.5 | 42.2 | 77.6 | 91.7 |
| Long-CLIP L/14 | 62.7 | 88.8 | 95.7 | 60.3 | 88.8 | 94.9 | 65.6 | 90.9 | 96.0 | 61.0 | 88.3 | 94.4 | 33.2 | 68.8 | 81.6 | 34.5 | 71.9 | 86.6 |
| InternVideo2$_{stage2}$ 1B | 72.5 | 93.7 | 97.3 | 69.5 | 94.6 | 97.8 | 72.4 | 94.2 | 97.4 | 62.7 | 90.5 | 95.9 | 46.0 | 80.8 | 91.9 | 46.6 | 82.5 | 92.5 |
| *LMM-based Models* | | | | | | | | | | | | | | | | | | |
| LLaVA-NV 7B[†] | 66.9 | 89.4 | 96.0 | 62.7 | 89.2 | 95.4 | 68.0 | 92.0 | 96.2 | 65.0 | 90.0 | 95.9 | 43.3 | 76.9 | 88.9 | 40.1 | 75.4 | 88.7 |
| MiniCPM-V 2.6[†] | 71.0 | 92.2 | 97.0 | 69.3 | 92.8 | 97.1 | 71.7 | 93.6 | 98.0 | 67.6 | 92.3 | 97.7 | 50.5 | 82.9 | 92.1 | 46.1 | 80.9 | 93.3 |
| InternVL2 8B[†] | 72.1 | 92.6 | 96.8 | 73.6 | 93.4 | 97.4 | 76.1 | 94.1 | 97.6 | 74.3 | 94.5 | 97.6 | 48.1 | 76.8 | 89.0 | 47.6 | 78.2 | 90.3 |
| Tarsier 7B[†] | 71.0 | 93.8 | 97.8 | 70.6 | 94.2 | 98.0 | 70.2 | 94.0 | 98.2 | 67.4 | 93.5 | 97.4 | 50.1 | 84.1 | 92.8 | 50.0 | 84.7 | 94.9 |
| Qwen2-VL 7B[†] | 76.6 | 95.3 | 98.7 | 77.4 | 95.6 | 98.7 | 78.2 | 95.5 | 98.5 | 75.4 | 95.0 | 98.1 | 51.9 | 84.8 | 94.9 | 52.7 | 85.4 | 95.2 |
| CaRe 7B | 77.0 | 95.6 | 98.7 | 79.0 | 96.8 | 99.1 | 76.8 | 96.3 | 98.7 | 78.1 | 95.8 | 99.3 | 50.7 | 85.3 | 94.4 | 53.4 | 86.3 | 94.0 |
| UNITE$_{base}$ 2B | 78.1 | 95.5 | 97.8 | 80.8 | 96.4 | 98.7 | 79.6 | 95.4 | 98.3 | 78.0 | 95.4 | 97.9 | 45.3 | 77.6 | 89.9 | 50.0 | 83.6 | 92.6 |
| UNITE$_{base}$ 7B | 86.0 | 96.8 | 98.9 | 86.9 | 98.3 | 99.7 | 86.5 | 96.9 | 99.2 | 84.8 | 98.0 | 99.4 | 52.4 | 82.5 | 92.2 | 55.4 | 86.5 | 95.3 |

Table 16: Detailed results on the MMEB benchmark. We average the scores in each meta-task.

| Model | Per Meta-Task Score | | | | Average Score | | |
|---|---|---|---|---|---|---|---|
| | Classification | VQA | Retrieval | Grounding | IND | OOD | Overall |
| #Datasets → | 10 | 10 | 12 | 4 | 20 | 16 | 36 |
| *Zero-shot Setting* | | | | | | | |
| CLIP (Radford et al., 2021) | 42.8 | 9.1 | 53.0 | 51.8 | 37.1 | 38.7 | 37.8 |
| BLIP2 (Li et al., 2023) | 27.0 | 4.2 | 33.9 | 47.0 | 25.3 | 25.1 | 25.2 |
| SigLIP (Zhai et al., 2023) | 40.3 | 8.4 | 31.6 | 59.5 | 32.3 | 38.0 | 34.8 |
| OpenCLIP (Cherti et al., 2023) | 47.8 | 10.9 | 52.3 | 53.3 | 39.3 | 40.2 | 39.7 |
| Magiclens (Zhang et al., 2024c) | 38.8 | 8.3 | 35.4 | 26.0 | 31.0 | 23.7 | 27.8 |
| E5-V 8B (LLaVA-NeXT) (Jiang et al., 2024b) | 21.8 | 4.9 | 11.5 | 19.0 | 14.9 | 11.5 | 13.3 |
| MMRet-MLLM 7B (LLaVA-1.6) (Zhou et al., 2024) | 47.2 | 18.4 | 56.5 | 62.2 | 43.5 | 44.3 | 44.0 |
| mmE5 11B (Llama-3.2-Vision) (Chen et al., 2025) | 60.6 | 55.7 | 54.7 | 72.4 | 57.2 | 62.9 | 58.6 |
| *Partially Supervised Finetuning Setting (Finetuning on M-BEIR)* | | | | | | | |
| UniIR (BLIP_FF) (Wei et al., 2024) | 42.1 | 15.0 | 60.1 | 62.2 | 44.7 | 40.4 | 42.8 |
| UniIR (CLIP_SF) (Wei et al., 2024) | 70.5 | 16.2 | 61.8 | 65.3 | 47.1 | 41.7 | 44.7 |
| MM-Embed 7B (LLaVA-1.6) (Lin et al., 2024) | 48.1 | 32.3 | 63.8 | 57.8 | - | - | 50.0 |
| GME 2B (Qwen2-VL) (Zhang et al., 2024d) | 56.9 | 41.2 | 67.8 | 53.4 | - | - | 55.8 |
| *Supervised Finetuning Setting (Finetuning on MMEB)* | | | | | | | |
| CLIP (Radford et al., 2021) | 55.2 | 19.7 | 53.2 | 62.2 | 47.6 | 42.8 | 45.4 |
| OpenCLIP (Cherti et al., 2023) | 56.0 | 21.9 | 55.4 | 64.1 | 50.5 | 43.1 | 47.2 |
| E5-V 4.2B (Phi3.5-V) (Jiang et al., 2024b) | 39.1 | 9.6 | 38.0 | 57.6 | 33.1 | 31.9 | 32.6 |
| E5-V 7B (LLaVA-1.6) (Jiang et al., 2024b) | 39.7 | 10.8 | 39.4 | 60.2 | 34.2 | 33.9 | 33.9 |
| VLM2Vec 4.2B (Phi-3.5-V) (Jiang et al., 2024c) | 54.8 | 54.9 | 62.3 | 79.5 | 66.5 | 52.0 | 60.1 |
| VLM2Vec 7B (LLaVA-1.6) (Jiang et al., 2024c) | 61.2 | 49.9 | 67.4 | 86.1 | 67.5 | 57.1 | 62.9 |
| VLM2Vec 2B (Qwen2-VL) (Jiang et al., 2024c) | 59.0 | 49.4 | 65.4 | 73.4 | 66.0 | 52.6 | 60.1 |
| VLM2Vec 7B (Qwen2-VL) (Jiang et al., 2024c) | 62.6 | 57.8 | 69.9 | 81.7 | 72.2 | 57.8 | 65.8 |
| MMRet-MLLM 7B (LLaVA-1.6) (Zhou et al., 2024) | 56.0 | 57.4 | 69.9 | 83.6 | 68.0 | 59.1 | 64.1 |
| UniME 4.2B (Phi3.5-V) (Gu et al., 2025) | 54.8 | 55.9 | 64.5 | 81.8 | 68.2 | 52.7 | 64.2 |
| UniME 7B (LLaVA-1.6) (Gu et al., 2025) | 60.6 | 52.9 | 67.9 | 85.1 | 68.4 | 57.9 | 66.6 |
| CAFe 0.5B (LLaVA-OneVision) (Yu et al., 2025) | 59.1 | 49.1 | 61.0 | 83.0 | 64.3 | 53.7 | 59.6 |
| CAFe 7B (LLaVA-OneVision) (Yu et al., 2025) | 65.2 | **65.6** | 70.0 | **91.2** | **75.8** | 62.4 | 69.8 |
| mmE5 11B (Llama-3.2-Vision) (Chen et al., 2025) | 67.6 | 62.8 | 70.9 | 89.7 | 72.3 | **66.7** | 69.8 |
| IDMR 8B (InternVL2.5) (Liu et al., 2025) | 58.3 | 58.6 | 68.7 | 85.6 | 70.5 | 57.9 | 64.9 |
| IDMR 26B (InternVL2.5) (Liu et al., 2025) | 66.3 | 61.9 | 71.1 | 88.6 | 73.4 | 63.9 | 69.2 |
| UNITE$_{instruct}$ 2B (Qwen2-VL) | 63.2 | 55.9 | 65.4 | 75.6 | 65.8 | 60.1 | 63.3 |
| UNITE$_{instruct}$ 7B (Qwen2-VL) | **68.3** | 65.1 | **71.6** | 84.8 | 73.6 | 66.3 | **70.3** |

Table 17: Detailed results of zero-shot performance on MSR-VTT (Xu et al., 2016), MSVD (Chen & Dolan, 2011), and DiDeMo (Anne Hendricks et al., 2017).

| Model | MSR-VTT | | | | | | MSVD | | | | | | DiDeMo | | | | | |
|---|---|---|---|---|---|---|---|---|---|---|---|---|---|---|---|---|---|---|
| | T→V | | | V→T | | | T→V | | | V→T | | | T→V | | | V→T | | |
| | R@1 | R@5 | R@10 | R@1 | R@5 | R@10 | R@1 | R@5 | R@10 | R@1 | R@5 | R@10 | R@1 | R@5 | R@10 | R@1 | R@5 | R@10 |
| *CLIP-based Models* | | | | | | | | | | | | | | | | | | |
| CLIP L/14 | 36.7 | 58.8 | 68.0 | 32.8 | 54.7 | 66.2 | 41.1 | 68.8 | 77.5 | 68.1 | 85.5 | 91.8 | 24.1 | 48.0 | 58.2 | 23.8 | 44.9 | 54.0 |
| Long-CLIP L/14 | 40.9 | 65.5 | 74.6 | 36.2 | 62.2 | 71.5 | 46.5 | 73.5 | 82.0 | 69.3 | 86.0 | 90.3 | 32.4 | 56.2 | 65.2 | 28.5 | 54.1 | 64.7 |
| InternVideo | 40.7 | - | - | 39.6 | - | - | 43.4 | - | - | 67.6 | - | - | 31.5 | - | - | 33.5 | - | - |
| LanguageBind | 42.1 | 65.9 | 75.5 | 40.1 | 65.4 | 73.9 | 50.0 | 77.7 | 85.6 | 75.1 | 90.0 | 94.2 | 35.6 | 63.6 | 71.7 | 35.6 | 62.8 | 71.8 |
| ViCLIP | 42.4 | - | - | 41.3 | - | - | 49.1 | - | - | 75.1 | - | - | 18.4 | - | - | 27.9 | - | - |
| *LMM-based Models* | | | | | | | | | | | | | | | | | | |
| VLM2Vec | 46.8 | 71.1 | 80.0 | - | - | - | 52.9 | 80.1 | 87.0 | - | - | - | - | - | - | - | - | - |
| LamRA 7B | 44.7 | 68.6 | 78.6 | - | - | - | 52.4 | 79.8 | 87.0 | - | - | - | - | - | - | - | - | - |
| CaRe 7B | 43.9 | 67.0 | 75.7 | 41.7 | 68.1 | 76.2 | 52.6 | 79.2 | 86.6 | 74.6 | 87.9 | 92.4 | 41.4 | 68.5 | 77.1 | 39.1 | 66.0 | 75.8 |
| UNITE$_{base}$ 2B | 43.8 | 67.6 | 76.8 | 41.7 | 66.2 | 75.9 | 50.0 | 77.8 | 85.1 | 73.1 | 90.7 | 94.3 | 37.9 | 64.1 | 73.7 | 26.4 | 37.5 | 63.5 |
| UNITE$_{base}$ 7B | 46.5 | 69.4 | 78.0 | 45.2 | 70.3 | 79.3 | 50.4 | 78.2 | 86.4 | 76.1 | 91.3 | 94.6 | 43.5 | 69.6 | 77.3 | 40.3 | 68.7 | 78.1 |

Table 18: Impact study of training data composition on general retrieval in the retrieval adaptation stage. We report zero-shot cross-modal retrieval results on coarse-grained cross-modal datasets (Flickr30K (Plummer et al., 2015), MSCOCO (Lin et al., 2014), MSR-VTT (Xu et al., 2016), MSVD (Chen & Dolan, 2011)) and fine-grained cross-modal datasets (DOCCI (Onoe et al., 2024), CaReBench-General(Xu et al., 2024b)) with Recall@1.

| Setting | | | Coarse Image-Text | | | | Coarse Video-Text | | | | Fine Image-Text | | Fine Video-Text | | Avg |
|---|---|---|---|---|---|---|---|---|---|---|---|---|---|---|---|
| | | | Flickr30K | | COCO | | MSR-VTT | | MSVD | | DOCCI | | CaRe-General | | |
| TT | TI | TV | T→I | I→T | T→I | I→T | T→V | V→T | T→V | V→T | T→I | I→T | T→V | V→T | |
| ✓ | | | 66.1 | 78.2 | 40.8 | 49.9 | 32.9 | 31.5 | 42.7 | 60.1 | 69.1 | 65.4 | 45.5 | 52.4 | 52.9 |
| | ✓ | | 68.1 | 79.5 | 42.6 | 58.2 | 37.2 | 35.6 | 43.7 | 66.4 | 75.8 | 71.8 | 57.9 | 62.9 | 58.3 |
| | | ✓ | **73.4** | **86.9** | **47.0** | **60.6** | **41.5** | **41.7** | **47.0** | **70.3** | **79.8** | **74.9** | **65.8** | **68.7** | **63.1** |
| ✓ | ✓ | | 67.7 | 78.9 | 43.5 | 56.3 | 38.5 | 35.2 | 44.6 | 66.3 | 74.8 | 70.3 | 56.8 | 58.8 | 57.6 |
| ✓ | | ✓ | 70.9 | 82.0 | 42.8 | 48.2 | 40.4 | 39.5 | 46.4 | 69.7 | 76.3 | 70.1 | 62.3 | 64.1 | 59.4 |
| | ✓ | ✓ | 71.6 | 84.2 | 45.0 | 59.3 | 41.3 | 41.4 | 46.0 | 70.3 | 77.8 | 73.0 | 65.7 | 68.4 | 62.0 |
| ✓ | ✓ | ✓ | 71.7 | 82.0 | 44.4 | 53.6 | 40.0 | 39.2 | 46.0 | 69.9 | 76.5 | 70.9 | 61.3 | 61.2 | 59.7 |

Table 19: Results of diverse training data composition on instruction-based retrieval in the retrieval adaptation stage.

| Setting | | | MMEB | | | | | | | WebVid-CoVR | | |
|---|---|---|---|---|---|---|---|---|---|---|---|---|
| TT | TI | TV | Classification | VQA | Retrieval | Grounding | IND | OOD | Overall | R@1 | R@5 | R@10 |
| ✓ | | | 60.9 | 53.8 | 61.1 | 73.7 | 61.9 | 58.5 | 60.4 | 64.4 | 86.3 | 92.1 |
| | ✓ | | 60.5 | 55.2 | 61.2 | 73.3 | 62.6 | 58.3 | 60.7 | **66.5** | 87.2 | 92.3 |
| | | ✓ | 60.9 | 55.0 | 61.9 | 74.1 | 62.6 | 59.1 | 61.1 | 65.6 | **87.8** | 92.4 |
| ✓ | ✓ | | **61.9** | 55.4 | **62.8** | **77.1** | **63.8** | **59.9** | **62.1** | 65.4 | 87.0 | 92.1 |
| ✓ | | ✓ | 59.7 | 54.4 | 60.7 | 74.2 | 62.1 | 57.7 | 60.2 | 65.6 | 87.4 | 92.6 |
| | ✓ | ✓ | 61.5 | **55.5** | 61.6 | 74.3 | 63.0 | 59.2 | 61.3 | 65.8 | 87.4 | 92.6 |
| ✓ | ✓ | ✓ | 61.2 | 54.7 | 61.8 | 74.6 | 62.7 | 59.0 | 61.0 | 64.8 | 86.8 | **92.7** |

## D CASE STUDY

Figure 7 illustrates visual examples of fine-grained retrieval results of UNITE 7B. UNITE demonstrates its ability to distinguish highly similar candidates, effectively identifying the most relevant items from visually and contextually similar items. For example, UNITE accurately recognizes location names (e.g., "majestic 25 de Abril Bridge in Lisbon, Portugal" in item 1) and textual content (e.g., "1856" in item 2 and "TATTOO" in item 3). It also identifies subtle differences (e.g., "cylindrical advertisement kiosk" in item 3). Additionally, UNITE excels at understanding temporal dynamics in videos, as shown by its precise object recognition in different scenes (items 4 and 5) and its ability to accurately discern temporal dynamic events in videos (item 6).

Figure 8 illustrates visual examples of instruction-based retrieval results of UNITE 7B. We sequentially present the results of four tasks in MMEB (Classification, VQA, Retrieval, and Visual Grounding) along with two results from WebVid-CoVR. UNITE proves effective even on challenging samples. For example, the second item requires precise recognition and understanding of tabular content, followed by reasoning to select the correct answer. In the last item, the input video and the top-3 candidates are highly similar, with the only difference being a small textual variation in one region. UNITE accurately identifies the video with the "banking" keyword, matching the background required by the textual instruction.

## E THE USE OF LARGE LANGUAGE MODELS

This section describes our use of LLMs for writing. Specifically, we employed LLMs to polish portions of our manuscript and to help check grammar.

LIMITATIONS

While UNITE demonstrates superior performance across text, image, and video modalities, audio emerges as another potential modality with the evolution of social media. Our exploration reveals that balancing multiple modalities remains challenging, suggesting the need for further investigation into modality expansion. Additionally, while comprehensive benchmarks exist for image-text retrieval, developing a unified benchmark that encompasses text, image, video, and potentially audio modalities represents a valuable future direction.

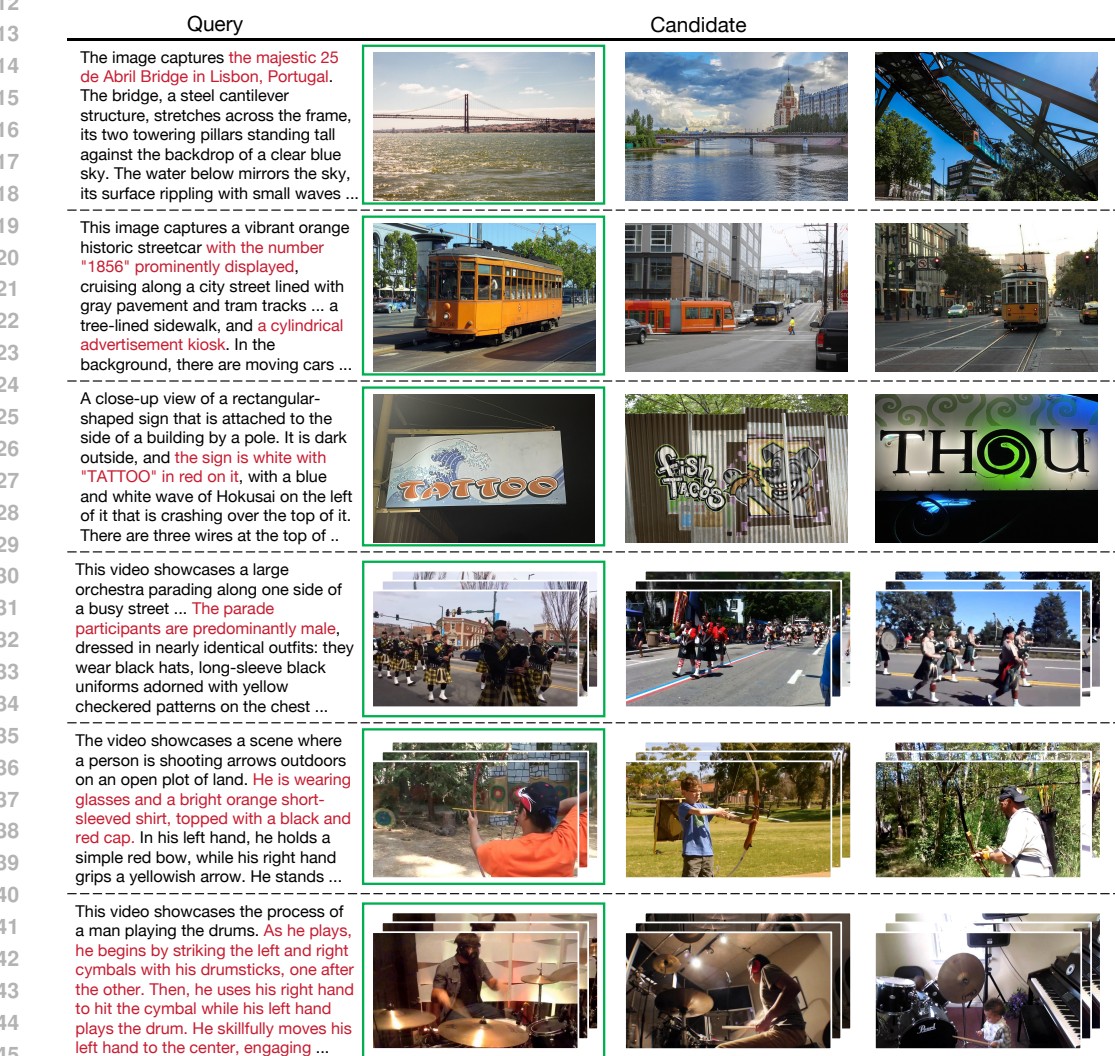

Figure 7: Visualization of fine-grained cross-modal retrieval results across ShareGPT4V, Urban1K, DOCCI, CaReBench-General, CaReBench-Spatial, and CaReBench-Temporal.

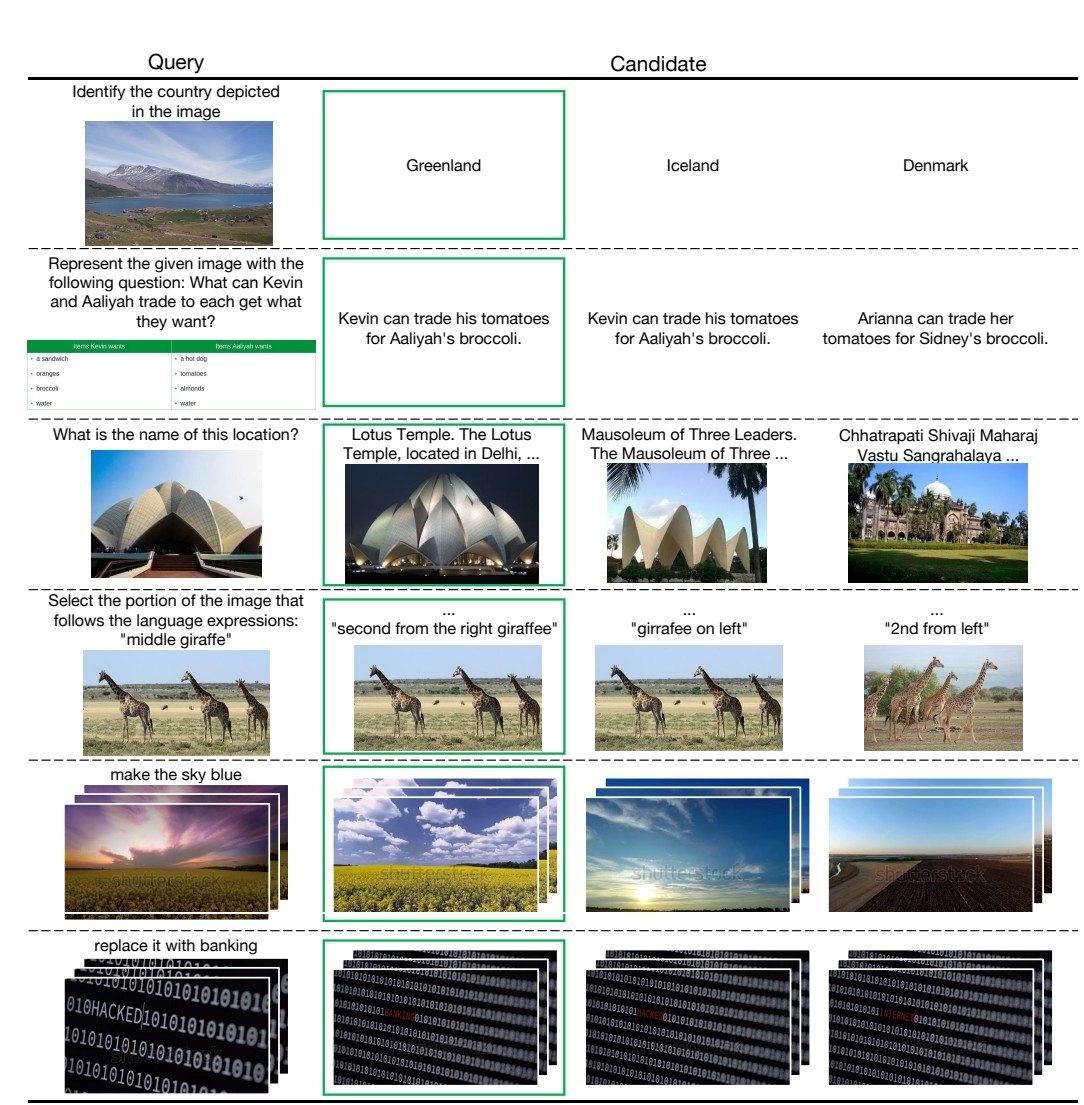

Figure 8: Visualization of instruction-based retrieval results across MMEB (Classification, VQA, Retrieval, and Visual Grounding tasks), and WebVid-CoVR.

Table 20: The detailed results of the baselines and our UNITE on MMEB, which includes 20 in-distribution (IND) datasets and 16 out-of-distribution (OOD) datasets. The out-of-distribution datasets are highlighted with a yellow background in the table. For each model, we report its best variant with fully available performance metrics, including VLM2Vec 7B (LLaVA-1.6) (Jiang et al., 2024c), MMRet 7B (LLaVA-1.6) (Zhou et al., 2024), UniME 7B (LLaVA-1.6) (Gu et al., 2025), mmE5 11B (Llama-3.2-Vision) (Chen et al., 2025), and IDMR 26B (InternVL2.5) (Liu et al., 2025). Superior versions might exist but are excluded due to incomplete score reporting.

| | CLIP | VLM2Vec | MMRet | UniME | mmE5 | IDMR | UNITE 2B | UNITE 7B |
|---|---|---|---|---|---|---|---|---|
| **Classification (10 tasks)** | | | | | | | | |
| ImageNet-1K | 55.8 | 74.5 | 58.8 | 71.3 | 77.8 | 80.6 | 77.6 | 80.2 |
| N24News | 34.7 | 80.3 | 71.3 | 79.5 | 81.7 | 81.6 | 66.8 | 80.3 |
| HatefulMemes | 51.1 | 67.9 | 53.7 | 64.6 | 64.2 | 72.3 | 57.4 | 67.1 |
| VOC2007 | 50.7 | 91.5 | 85.0 | 90.4 | 91.0 | 92.7 | 85.0 | 84.9 |
| SUN397 | 43.4 | 75.8 | 70.0 | 75.9 | 77.7 | 78.8 | 75.2 | 78.7 |
| Place365 | 28.5 | 44.0 | 43.0 | 45.6 | 43 | 38.9 | 41.3 | 44.5 |
| ImageNet-A | 25.5 | 43.6 | 36.1 | 45.5 | 56.3 | 63.6 | 48.3 | 59.2 |
| ImageNet-R | 75.6 | 79.8 | 71.6 | 78.4 | 86.3 | 84 | 88.8 | 90.5 |
| ObjectNet | 43.4 | 39.6 | 55.8 | 36.4 | 62.5 | 50.5 | 66.6 | 68.1 |
| Country-211 | 19.2 | 14.7 | 14.7 | 18.7 | 35.4 | 20.3 | 24.8 | 29.5 |
| *All Classification* | 42.8 | 61.2 | 56.0 | 60.6 | 67.6 | 66.3 | 63.2 | 68.3 |
| **VQA (10 tasks)** | | | | | | | | |
| OK-VQA | 7.5 | 69.0 | 73.3 | 68.3 | 67.6 | 71.0 | 57.3 | 67.1 |
| A-OKVQA | 3.8 | 54.4 | 56.7 | 58.7 | 56.1 | 59.2 | 46.0 | 58.0 |
| DocVQA | 4.0 | 52.0 | 78.5 | 67.6 | 90.3 | 75.1 | 87.0 | 92.7 |
| InfographicsVQA | 4.6 | 30.7 | 39.3 | 37.0 | 56.5 | 44.6 | 52.8 | 71.3 |
| ChartQA | 1.4 | 34.8 | 41.7 | 33.4 | 50.5 | 64.6 | 45.7 | 63.2 |
| Visual7W | 4.0 | 49.8 | 49.5 | 51.7 | 51.9 | 54.9 | 47.7 | 54.9 |
| ScienceQA | 9.4 | 42.1 | 45.2 | 40.5 | 55.8 | 54.7 | 41.1 | 51.2 |
| VizWiz | 8.2 | 43.0 | 51.7 | 42.7 | 52.8 | 47.1 | 49.8 | 53.4 |
| GQA | 41.3 | 61.2 | 59.0 | 63.6 | 61.7 | 71.0 | 53.0 | 56.8 |
| TextVQA | 7.0 | 62.0 | 79.0 | 65.2 | 83.3 | 77.0 | 78.9 | 82.3 |
| *All VQA* | 9.1 | 49.9 | 57.4 | 52.9 | 62.6 | 61.9 | 55.9 | 65.1 |
| **Retrieval (12 tasks)** | | | | | | | | |
| VisDial | 30.7 | 80.9 | 83.0 | 79.7 | 74.1 | 81.5 | 70.0 | 80.5 |
| CIRR | 12.6 | 49.9 | 61.4 | 52.2 | 54.7 | 57.6 | 43.5 | 51.6 |
| VisualNews_t2i | 78.9 | 75.4 | 74.2 | 74.8 | 77.6 | 78.5 | 70.6 | 79.3 |
| VisualNews_i2t | 79.6 | 80.0 | 78.1 | 78.8 | 83.3 | 80.6 | 74.1 | 82.4 |
| MSCOCO_t2i | 59.5 | 75.7 | 78.6 | 74.9 | 76.4 | 79.1 | 73.6 | 78.2 |
| MSCOCO_i2t | 57.7 | 73.1 | 72.4 | 73.8 | 73.2 | 75.4 | 71.1 | 74.3 |
| NIGHTS | 60.4 | 65.5 | 68.3 | 66.2 | 68.3 | 68.6 | 65.1 | 66.0 |
| WebQA | 67.5 | 87.6 | 90.2 | 89.8 | 88.0 | 89.0 | 85.0 | 87.0 |
| FashionIQ | 11.4 | 16.2 | 54.9 | 16.5 | 28.8 | 21.0 | 18.1 | 26.3 |
| Wiki-SS-NQ | 55.0 | 60.2 | 24.9 | 66.6 | 65.8 | 66.9 | 65.0 | 72.2 |
| OVEN | 41.1 | 56.5 | 87.5 | 55.7 | 77.5 | 67.4 | 65.9 | 73.1 |
| EDIS | 81.0 | 87.8 | 65.6 | 86.2 | 83.7 | 87.6 | 82.2 | 88.3 |
| *All Retrieval* | 53.0 | 67.4 | 69.9 | 67.9 | 71.0 | 71.1 | 65.4 | 71.6 |
| **Visual Grounding (4 tasks)** | | | | | | | | |
| MSCOCO | 33.8 | 80.6 | 76.8 | 76.5 | 53.7 | 81.5 | 64.6 | 73.9 |
| RefCOCO | 56.9 | 88.7 | 89.8 | 89.3 | 92.7 | 91.7 | 79.5 | 89.2 |
| RefCOCO-matching | 61.3 | 84.0 | 90.6 | 90.6 | 88.8 | 88.1 | 84.5 | 90.1 |
| Visual7W-pointing | 55.1 | 90.9 | 77.0 | 84.1 | 92.3 | 93.1 | 73.9 | 86.1 |
| *All Visual Grounding* | 51.8 | 86.1 | 83.6 | 85.1 | 89.6 | 88.6 | 75.6 | 84.8 |
| **Final Score (36 tasks)** | | | | | | | | |
| All | 37.8 | 62.9 | 64.1 | 66.6 | 69.8 | 69.2 | 63.3 | 70.3 |
| All IND | 37.1 | 67.5 | 59.1 | 68.4 | 72.3 | 73.4 | 65.8 | 73.6 |
| All OOD | 38.7 | 57.1 | 68.0 | 57.9 | 66.7 | 63.4 | 60.1 | 66.3 |

