# OpenReview forum: "Modality Curation: Building Universal Embeddings for Advanced Multimodal Information Retrieval"
_ICLR.cc/2026/Conference — Submitted to ICLR 2026_

### Official Review · Reviewer_MRmf · 2025-10-29

**Soundness:** 2
**Presentation:** 3
**Contribution:** 2
**Rating:** 4
**Confidence:** 5

**Summary:**

This paper addresses modality heterogeneity and cross-modal alignment in Multimodal Information Retrieval (MIR) by proposing UNITE, a universal embedding framework with two core innovations: systematic modality curation and Modal-Aware Masked Contrastive Learning (MAMCL). Trained via a two-stage pipeline, UNITE achieves SOTA on 40+ tasks and supports simultaneous fine-grained/instruction-based retrieval for text/image/video/fusions.

**Strengths:**

1.	UNITE achieves leading results across 40+ retrieval tasks (coarse-grained, fine-grained, instruction-based), outperforming both smaller specialized models and larger competitors (e.g., 2B UNITE surpasses 7B VLM2Vec on WebVid-CoVR).
2.	Systematic analysis reveals T-V pairs excel in general retrieval and even outperform T-I pairs in image-text tasks—contradicting traditional assumptions and guiding more efficient MIR data curation. It's a interesting finding.

**Weaknesses:**

1.	Inadequate Theoretical and Modal Coverage in Modality Curation. The paper focuses on analysis of T-V pair effectiveness but lacks theoretical explanations for why certain data types (e.g., T-V outperforming T-I in image-text tasks) yield better results. Additionally, while it emphasizes "curating modality data" as a core contribution, the analysis is mostly limited to T→V retrieval conclusions. It fails to clarify how other modalities (e.g., text-text, image-text) should be managed—whether they are still mixed as in traditional methods, or if there are optimized proportion strategies?
2.	MAMCL’s design lacks sufficient novelty, as prior contrastive learning works (e.g., sampling batches from a single data source to ensure uniform modality) have already addressed cross-modality interference, making MAMCL a similar but not groundbreaking approach. Moreover, MAMCL masks negative examples of different modalities, which wastes a portion of negative samples. Given that the quantity and quality of negative examples are critical for contrastive learning performance, this waste may limit the model’s ability to learn discriminative representations.

**Questions:**

1.	Table 7 (1→2) shows that MAMCL reduces the performance of WebVid CoVR. Does clarifying this mean that MAMCL will cause losses in specific circumstances?
2.	What other functions does the conclusion obtained in Analysis 5 have, apart from guiding the addition of T-V data in retrieval adaptation?  Is the data- curation meaningless, merely indicating that the data of TV needs to be included? Because in the end, all the data were mixed together, just like the previous work. Moreover, the proportion of different modal data has not been explored either.
3.	The article does not provide specific prompts for Instruction-based Retrieval. Since instructions determine the tuning results, omitting them will hinder replication. Supplement these in appendix.

---

> ### Author Response · Authors · 2025-11-20
> **Response to Reviewer MRmf Part 1**
>
> We sincerely appreciate the reviewer’s thoughtful comments and valuable suggestions. The reviewer recognizes the strong performance of our model and expresses interest in our key findings. We have addressed the reviewer’s concerns and questions in detail, supported by additional experimental results. We hope our responses clarify the reviewer’s concerns. The paper has been improved based on these feedback, and we believe our discussion will further enhance the work, providing effective insights for the community.
>
> ---
>
> ### Q1: Inadequate Theoretical and Modal Coverage in Modality Curation
>
> We sincerely appreciate the reviewer for raising the important concern about theoretical depth and modality curation. Your feedback has helped us further clarify related contents.
>
> The observation that T-V outperforms T-I in image–text retrieval tasks can be explained as follows:
>
> 1. Videos provide richer and more dynamic visual context than images. Videos involve motion, event progression, and temporal evolution, enabling the model to learn broader and more generalizable vision–language alignment. Moreover, an image can be regarded as a special case of a video (i.e., a single static frame) [1]. Therefore, once the model has learned to align complex video–text semantics, transferring to static images, where semantics are simple, can yield superior performance.
> 2. Video–text training introduces harder negatives, improving discriminative ability. Due to temporal complexity, video–text contrastive learning often encounters more challenging negative samples (e.g., differing action sequences). This forces the model to learn stronger semantic discrimination. When transferred to image–text tasks, this enhanced discriminative ability helps distinguish finer semantic differences, improving performance.
>
> Regarding the reviewer's question on how other modalities (e.g., text–text, image–text) are handled, our analysis in L422–431 already summarizes their roles. As shown in Tab. 8, although TV is the most effective for general cross-modal retrieval, TV alone performs suboptimally on instruction-based retrieval, where TT + TI yields the best results. Therefore, TT and TI are indispensable for comprehensive retrieval capability. Specifically:
> - TT enhances linguistic understanding and logical reasoning, which are critical for instruction-following tasks;
> - TI provides tighter semantic alignment between vision and language, offering more precise multimodal correspondence.
>
> Tab. 8 (first 8 columns) shows performance of different data compositions on coarse-grained and fine-grained cross-modal retrieval (I-T and V-T), where TV-only is the best. The following table further list results by input–output modality patterns to reveal the roles of TT and TI more clearly:
>
> | TT | TI | TV | I→T | T→I | I→I | T+I→T | T+I→I | T→T+I | T+I→T+I | T+V→V | Avg. |
> | - | - | - |  -   |  -   |  -   |  -   |  -   |  -   |  -   |  -   |  -   |
> | ✓ |   |   | 61.7 | 68.7 | 66.4 | 53.8 | 63.4 | 78.4 | 40.1 | 64.4 | 62.1 |
> |   | ✓ |   | 61.4 | 70.1 | 66.0 | 55.2 | 63.6 | 75.5 | 39.0 | 66.5 | 62.2 |
> |   |   | ✓ | 61.9 | 69.6 | 65.9 | 55.0 | 64.7 | 78.0 | 39.7 | 65.6 | 62.5 |
> | ✓ | ✓ |   | 62.8 | 70.1 | 65.2 | 55.4 | 66.5 | 77.3 | 45.6 | 65.4 | 63.5 |
> | ✓ |   | ✓ | 60.4 | 68.0 | 64.1 | 54.4 | 64.0 | 78.4 | 40.4 | 65.6 | 61.9 |
> |   | ✓ | ✓ | 62.3 | 69.5 | 65.4 | 55.5 | 64.8 | 76.4 | 40.5 | 65.8 | 62.5 |
> | ✓ | ✓ | ✓ | 62.0 | 69.5 | 65.2 | 54.7 | 64.1 | 78.1 | 42.5 | 64.8 | 62.6 |
>
> TV shows a slight lead for single-modality training. However, TT + TI is the optimal combination for instruction-based retrieval, with TT + TI + TV next, confirming the importance of TT and TI. We also observe a consistent pattern: **settings involving TI outperform those involving TT (TI > TT and TI+TV > TT+TV)**. Thus, we intentionally allocate more TI than TT in our pretraining data composition (TI: 39.2% vs. TT: 21.6%), while keeping a large proportion of TV (36.1%) due to its strength in general cross-modal retrieval, resulting in a balanced configuration for overall performance. **Our final training data are obtained through sampling from designated datasets, rather than indiscriminately aggregating all available data.** *In contrast to MMRet/BGE-VL, which relies on 26M training samples, and IDMR, which uses a much larger 26B model, our data curation and method achieve stronger performance with less training data and a 7B model.*
>
> ---
> [1] Frozen in Time: A Joint Video and Image Encoder for End‑to‑End Retrieval, ICCV 2021

---

> > ### Comment · Reviewer_MRmf · 2025-11-26
> >
> > Thanks authors for the responses, which addressed some of my questions.
> >
> > ---
> > > Video–text training introduces harder negatives, improving discriminative ability. Due to temporal complexity, video–text contrastive learning often encounters more challenging negative samples (e.g., differing action sequences). This forces the model to learn stronger semantic discrimination. When transferred to image–text tasks, this enhanced discriminative ability helps distinguish finer semantic differences, improving performance.
> >
> > Could you provide some visual examples to support the statement?
> >
> > > the common approach of multi-task learning is to mix multi-task datasets with random sampling
> >
> > It's common choice for multi-task learning, but not for retrieval.
> >
> > And for the novelty part, the mask strategy is introduced in existing works, such as GISTEmbed (https://arxiv.org/pdf/2402.16829).
> >
> >
> > > We would be grateful if the reviewer could provide specific references for “sampling batches from a single data source”, and we would be happy to include a discussion of these works in our paper.
> >
> > It's a default implementation in early versions of `sentence_transformers`.
> > And it is stated in models such as GTE (https://arxiv.org/pdf/2308.03281, "we ensure that all training instances within a batch originate from the same task.")
> >
> > However, your new results is helpful to support your motivation.
> >
> > I still feel that your masking/filtering and the mixed-source batching are kind of conflicting. Samples from different modalities are mixed in the same batch, but they don’t actually interact with each other. They could just be separated instead, which would give you a larger space for in-batch negatives.

---

> ### Author Response · Authors · 2025-11-20
> **Response to Reviewer MRmf Part 2**
>
> ### Q2: MAMCL's design lacks sufficient novelty.
>
> We sincerely appreciate the reviewer for this thoughtful feedback. We would address each concern below.
>
> #### 1. Difference from *sampling batches from a single data source*
>
> We would be grateful if the reviewer could provide specific references for “sampling batches from a single data source”, and we would be happy to include a discussion of these works in our paper.
>
> While the motivation is similar, **MAMCL is fundamentally different in both mechanism and scope**. Single-source sampling is a pure data-level solution, whereas MAMCL operates at the training loss level. In addition, based on multi-task training practices, concurrent works, and our supplementary experiments, we argue that “single-source batch sampling” is not a sufficient solution for universal multimodal retrieval.
>
> - **The common approach of multi-task learning is to mix multi-task datasets with random sampling [2–4], which benefits zero-shot generalization.** These works encourage unified training with mixed data and random sampling, rather than single-source batches. Furthermore, a common observation is that later training data tends to overwrite parameters learned from earlier task, causing bias toward later sources. With single-source batches, models favor sources appearing later, potentially degrading earlier task performance. In extreme cases, tasks with fewer data suffer severe performance loss, while tasks with oversized data may dominate at the expense of others.
> - Recent work [5] further highlights this trend: “interleaving multiple such sub-batches preserves cross-task diversity within the full batch, avoiding **the instability commonly observed in completely homogeneous batches that originate from a single source.**” This indicates that single-source batch sampling often causes training instability, and that mixed-source batches is preferred.
> - To further explore the difference, we conducted experiments with minimal cost (given our resource and time constraints). We train and test on retrieval tasks from MMEB-Train (8 tasks, 15K samples each, 120K total) and evaluate on all 12 MMEB-Eval retrieval tasks. Randomly shuffled mixed-source sampling as the baseline. For single-source sampling, each batch contains samples from only one source, while the batch order is randomized.
>
> |  | mixed-source | single-source | mixed + MAMCL |
> |-|:-:|:-:|:-:|
> | Retrieval Tasks | 66.0 | 65.6 | 66.8 |
>
> Experiments show that single-source sampling leads to training instability. Even so, during our recent investigation, we notice that the newly released EBind [6] explores single-source sampling in another setting (different from ours) and demonstrates promising potential. We believe this strategy can be effective in certain scenarios. However, under our setup, it does not outperform our MAMCL, and we look forward to future studies on this technique.
>
> In summary, MAMCL differs fundamentally from prior approaches and is both theoretically sound and empirically effective. **When facing real-world imbalanced data distributions, more modalities, and diverse tasks, we believe MAMCL will prove more stable and effective than prior methods.**
>
> #### 2. MAMCL masks negative examples of different modalities
>
> We agree that MAMCL discards a portion of negative samples. However, the masked negatives are not necessarily useful. These samples are typically trivial negatives that the model can easily identify as irrelevant and thus contribute minimally to contrastive learning. Recent works [7] demonstrates that **blindly increasing the number of negatives brings diminishing returns and can even introduce noise**. Effective contrastive learning benefits from semantically related yet non-conflicting negatives, not simply more negatives. Moreover, MAMCL still employs mixed-source sampling within batches, using masking to retain **high-quality negatives** consistent with the current target modality.
>
> ---
> [2] Scaling Instruction-Finetuned Language Models, JMLR 2024
> [3] The Flan Collection: Designing Data and Methods for Effective Instruction Tuning, ICML 2023
> [4] Dynamic Data Mixing Maximizes Instruction Tuning for Mixture-of-Experts, NAACL 2025
> [5] VLM2Vec-V2: Advancing Multimodal Embedding for Videos, Images, and Visual Documents, arXiv:2507.04590 (Technical Report)
> [6] EBind: a practical approach to space binding, arXiv:2511.14229
> [7] Breaking the Batch Barrier (B3) of Contrastive Learning via Smart Batch Mining, NeurIPS 2025

---

> ### Author Response · Authors · 2025-11-20
> **Response to Reviewer MRmf Part 3**
>
> ### Q3: Table 7 (1→2) shows that MAMCL reduces the performance of WebVid CoVR. Does clarifying this mean that MAMCL will cause losses in specific circumstances?
>
> We sincerely appreciate the reviewer for this meaningful comment.
>
> We clarify a potential misunderstanding: ID 1→2 compares MMEB performance with/without MAMCL (WebVid-CoVR is not trained here). MMEB contains many modality combinations (e.g., T→I, T+I→T, T+I→T+I; see Tab. 11), which can lead to competitive interactions among modalities. With MAMCL applied, performance on MMEB clearly improves, demonstrating its effectiveness in mitigating modality interference.
>
> **For WebVid-CoVR, the correct comparisons are (3–4–5).** ID 3→4: introducing MMEB reduces CoVR performance (69.2 → 67.4). ID 4→5: applying MAMCL alleviates cross-modal interference and restores CoVR performance to the level of dedicated training (67.4 → 69.1).
>
>
> ### Q4: What other functions does the conclusion obtained in Analysis 5 have, apart from guiding the addition of T-V data in retrieval adaptation?
>
> We sincerely appreciate the reviewer for this meaningful comment. It is possible that our writing has misled the reviewer into thinking that the conclusions in Sec. 5 are limited. We would like to clarify this below.
>
> The analysis in Sec. 5 is necessary because the community currently lacks a systematic empirical understanding of training with text, image, and video jointly across various input–output patterns. Our findings fill this gap. Beyond informing the addition of T-V data in retrieval adaptation, the analysis reveals the distinct roles of different modality types during pretraining. Specifically, T-V is the most critical for general cross-modal retrieval, while T-T and T-I are essential for instruction-based retrieval (as detailed in our response to Q1). Considerations on the optimal modality composition therefore naturally follow from these observations.
>
> Additionally, our data curation is not an optimization limited to specific benchmarks, but a generalizable methodology. In addition to the zero-shot and supervised results reported in the main paper, UNITE also performs strongly on a wide range of unseen benchmarks [8], iincluding CameraBench (CMRB), DREAM-1K-Event (DREAM-E), LoVR-Theme2Clip (LoVR-TH), LoVR-Text2Video (LoVR-V), LoVR-Clip-to-Video (LoVR-C2V), PE-Video-Keyword (PEV-K), and MSRVTT-ImageVideo (MSRVTT-I2V):
>
> | Method     | CMRB | DREAM-E | LoVR-TH | LoVR-V | LoVR-C2V | PEV-K | MSRVTT-I2V | Avg. |
> |-|:-:|:-:|:-:|:-:|:-:|:-:|:-:|:-:|
> |            | R@10 | R@1  | R@10 | R@1  | R@1  | R@1  | R@1  |      |
> | VLM2Vec-V2 | 28.6 | 22.8 | 49.2 | 61.0 | 38.5 | 32.4 | 84.1 | 45.2 |
> | GME-7B     | 30.4 | 27.4 | 52.3 | 71.0 | 37.0 | 39.6 | 86.0 | 49.1 |
> | Unite-7B   | 35.1 | 27.9 | 55.5 | 62.0 | 44.8 | 44.0 | 88.3 | 51.1 |
>
> The consistent gains across such diverse and previously unseen retrieval scenarios demonstrate that **our data curation contributes to the broader goal of universal multimodal retrieval.**
>
>
> ### Q5: The article does not provide specific prompts for Instruction-based Retrieval
>
> We sincerely appreciate the reviewer for pointing out the need for clearer details on our prompts.
>
> The prompt used in our work is provided in L161: `<vision>\n<text>\nSummarize above <modalities> in one word:`, with minor variations described in L179–180. A prompt example is also shown in Fig. 2(a). Since our code is open-sourced, the exact prompts used in training and inference can be easily accessed.
>
> For clarity, here are explicit prompt forms for partial query types:
> - Text query: `<text>\nSummarize above sentence in one word:`
> - Image query: `<vision>\nSummarize above image in one word:`
> - Text + Video query: `<vision>\n<text>\nSummarize above sentence and video in one word:`
>
> Instruction-based retrieval samples follow the prompt template above and retain the original semantics of the dataset. For instance, an MMEB query: `<|image_1|>\nRepresent the given image with the following question: What is the man by the bags awaiting?` becomes in Unite: `<vision>\nRepresent the given image with the following question: What is the man by the bags awaiting?\nSummarize above sentence and image in one word:`
>
> In summary, our prompts are unified, already shown in the paper, and the required modifications are straightforward. **Notably, because our training code is fully open-sourced, both prompts and the whole training are transparent and fully reproducible.**
>
> ---
>
> [8] Towards Universal Video Retrieval: Generalizing Video Embedding via Synthesized Multimodal Pyramid Curriculum, arXiv:2510.27571
>
> ---
>
> We sincerely appreciate the reviewer’s thoughtful and constructive suggestions. We have greatly benefited from these comments and believe that they will help make our paper even stronger.

---

> ### Author Response · Authors · 2025-11-26
> **Additional Response to Reviewer MRmf**
>
> Thank you very much for taking the time to read and respond to our rebuttal.
>
> ---
>
> > Could you provide some visual examples to support the statement?
>
> For simplicity, we refer to the examples in Fig. 7/8 of the paper. Although the examples comes from benchmarks, similar patterns also appear in our training data. First, videos contain unique temporal nature, requiring models to capture dynamic temporal information rather than purely static elements. Second, describing a video typically requires more content, while the core distinguishing cues may be proportionally much smaller. This makes video-text alignment more challenging than image–text alignment.
>
> For example, in the last row of Fig. 7, text is temporal description. The first two candidates share extremely similar static visual information (people, scene) and differ only in action sequence. Distinguishing the correct candidate requires stronger visual understanding. In the last row of Fig. 8, the three candidates differ only by a tiny piece of text that occurs in just 2–3 frames, which forces the model to locate and utilize highly localized signals.
>
> ---
>
> > It's common choice for multi-task learning, but not for retrieval. And for the novelty part, the mask strategy is introduced in existing works, such as GISTEmbed.
>
> We sincerely appreciate the reviewer for pointing out GISTEmbed as a potentially related work. After carefully reading, we believe that MAMCL is fundamentally different from the masking strategy used in GISTEmbed.
>
> - GISTEmbed: Its masking targets **false negatives**. That is, when a candidate negative is highly similar to a positive, treating it as a negative would mislead the model. Their goal is to mask such likely relevant negatives to avoid representational collapse. As stated in the original paper: “some randomly selected text in another triplet within the same batch may be relevant to the query-positive pair being evaluated. If we use such examples as negatives, the model will likely learn a suboptimal representation of the embeddings” and “masking the likely ‘relevant’ samples with $−\infty$, ensuring they will have no contribution to the loss.”
> - our MAMCL: Its goal is to **alleviate cross-modal competition**. The masking is performed depending on the **target modality**, not on semantic relevance.
>
> Therefore, the masking criteria and the motivation are essentially different, meaning that the existence of GISTEmbed does not affect the novelty of MAMCL.
>
> We hope this clear comparison helps convey the conceptual differences. We would also be happy to discuss these distinctions explicitly in the paper to better assist readers.
>
> ---
>
> > It's a default implementation in early versions of `sentence_transformers`. And it is stated in models such as GTE
> > Samples from different modalities are mixed in the same batch, but they don’t actually interact with each other. They could just be separated instead, which would give you a larger space for in-batch negatives.
>
> We sincerely appreciate the reviewer for pointing out the implementation in `sentence_transformers` and GTE. This observation is valuable, and we agree that single-source batch sampling can be a reasonable choice in certain settings. We would like to clarify the difference in applicability.
>
> - **sentence_transformers and GTE operate in text-only domain**, where all samples share the same modality. In that setting, single-source batching is suitable for multi-task learning in text-only retrieval.
> - **MAMCL focuses heterogeneous modalities (text, image, video, and their combinations)** with different target alignment objectives. MAMCL is specifically designed to handle modality interference in this setting.
>
> Thus, the core distinction is that GTE solves **task heterogeneity under a fixed modality**, while our method addresses **modality heterogeneity with multiple target alignment objectives**. Applying single-source batching directly to multimodal retrieval often leads to batches dominated by a single modality, which can overwrite previously learned modality-specific alignment and cause systematic bias against weaker modalities.
>
> Regarding the concern that masking may waste negative samples, we agree that masking discards some low-value negatives. We do not deny the effectiveness of single-source batching and acknowledge that it can be valuable in certain scenarios. Our core point is that **“across-batch” modality bias introduced by single-source batching is more harmful than the potential “within-batch” loss of negatives**. Recent influential work [5] also note that single-source batching can lead to unstable training. MAMCL effectively mitigates cross-modal interference and improves the usefulness of negatives within diverse mixed batches. Our experiments show that this balance leads to more stable and robust multimodal embedding.
>
> ---
>
> Thank you again for your timely response and constructive discussion. We hope the explanations address the your concerns. Have a nice day!

---

> > ### Comment · Reviewer_MRmf · 2025-11-27
> >
> > Thank you for clarifying that your masking objective! Sorry, it’s been a while, and I lost track of that detail.
> >
> > From what I understand, your masking is used to ensure that all negatives come from the same modality. But honestly, that feels even more trivial than the false negative issues. You could achieve exactly the same goal just by adjusting your batch sampling strategy, e.g., by constructing each batch using data from only one modality. This would work regardless of whether you’re using mix-source sampling or not.
> >
> > I think your masking seems to be compensating for a limitation in code your implementation.
> > If you use modality-specific batch sampling, you’d actually get more effective in-batch negatives, which means you can learn richer signals. You’d end up with a much denser similarity matrix than the sparse one shown in your Figure 2b, which honestly feels like a waste of valuable batch samples.
> >
> > For example, consider a batch with 8 queries, each paired with 1 positive & 2 negatives—this gives a query-document relevance matrix of size 8×24. Suppose your MAMCL only learns from 50% of the cells (which is actually more optimistic than your Figure 2b); that means each query sees only 12 in-batch negatives. In contrast, if you simply control the batch sampling (e.g., by ensuring all items in the batch come from the same modality), you could utilize all cells—giving each query access to 24 in-batch negatives. This is clearly the better choice, especially since it’s widely accepted that more negatives generally lead to better performance.
> >
> > Even setting performance aside, your method is less efficient computationally. To achieve the same number of in-batch negatives, say 12, your approach requires forwarding 8 × 4 = 32 input samples (8 queries + 24 docs), whereas simply adjusting the batching strategy only needs 8 × 2.5 = 20 samples (8 queries + 12 docs). Under identical training conditions, which is entirely achievable since the same training pairs can be sampled, your method incurs 37.5% more compute overhead for no added benefit.
> >
> > From this perspective, MAMCL has fairly significant shortcomings and lacks genuine novelty. I’m happy to acknowledge “using samples from the same modality in one batch” as your practical training strategy contribution, but as a method, MAMCL doesn’t bring any real advantages.
> >
> > Framing a strategy that can be easily implemented as an engineering trick as a method with serious flaws, in my view, doesn’t add any real value.
> >
> > ---
> >
> > Overall, your work proposes a simple data sampling (batching from heterogeneous modality data) and training strategy (video-text pairs is very effective) and reports a SOTA model as a result.
> >
> > To be honest, I’d consider this exactly a borderline work. And your average rating is above this level.
> > I sincerely appreciate the authors’ hard work and thorough responses.

---

> > > ### Author Response · Authors · 2025-11-27
> > > **Additional Response to Reviewer MRmf Part 3**
> > >
> > > We sincerely thank the reviewer for the follow up.
> > >
> > > Regarding the relation to single-source batching, our earlier explanation may not be sufficiently clear. In our recent response (*Additional Response to Reviewer MRmf Part 2*), we provided a additional view from the gradient dynamics. This should address some of your concerns, particularly your concern that our masking strategy is less effective than single-source batching. We believe MAMCL is not equivalent to single-source batching and cannot be replaced. We have also pointed out in our previous responses that the motivation behind our mask strategy is not unfounded, aligning with recent work [5,7].
> > >
> > > Regarding the concern about wasting negatives, with sufficiently large batch size, the number of remaining negatives per query is still enough, and MAMCL acts as a fewer but higher quality negative selection. If simply increasing the number of negatives is always optimal, the community can obtain better performance by expanding batch size to 8K or 16K, but this is not the case.
> > >
> > > Single-source batching has indeed been used in early `sentence_transformers` and in GTE for text-only retrieval. To the best of our knowledge, *this strategy has not been adopted as a standard practice in multimodal retrieval after all this time*. **Mixed-source batching remains the default in recent multimodal embedding systems, including the follow-up works from the GTE team such as GME and GVE, which do not apply single-source batching in the same way.** We believe this difference reflects a nature gap between text-only multi-task retrieval and multimodal retrieval, rather than a mere implementation artifact.

---

> ### Author Response · Authors · 2025-11-27
> **Additional Response to Reviewer MRmf Part 2**
>
> To supplement our explanation, we offer a comparison between single-source batching and mixed-modal batching from the perspective of training gradients.
>
> - Single-source batching: All gradients within a batch originate from a single modality, **updating parameters toward a modality-specific optimum**. When batches from different modalities, the gradient direction oscillates toward different optimum. For models such as GTE, which operate in text-only domain, the gap between tasks within the same modality is relatively small, so this oscillation is less. In contrast, when heterogeneous modalities are involved, the oscillation often leads to unstable training [5].
> - Mixed-modal batching: The model is **exposed to multiple modalities within each optimization step**. **The gradient update becomes a weighted combination of objectives across modalities instead of a modality-specific update.** This reduces across-batch variance in gradient direction and mitigates oscillation. Although MAMCL masks negatives that are irrelevant to the current target modality, **gradients from different modalities still exist within the batch** (see Fig. 2(b)), while unnecessary modality conflicts are filtered out.
>
> Therefore, MAMCL is well motivated and empirically effective. Additionally, to prevent other readers from encountering similar confusion, we will incorporate the above discussion into the next version of our paper, including a clearer comparison with these existing works (GISTEmbed, GTE, ...). We believe this will provide a more clearer distinction between our approach and prior methods.

---

> ### Comment · Reviewer_MRmf · 2025-11-27
>
> I’ve checked the GME and GVE papers, and neither explicitly specifies whether they use single-source or mixed-source sampling, so please don’t create a misleading impression. If I’ve overlooked anything, please point it out, I’m happy to be corrected.
>
> ---
>
> As I stated earlier, the core issue now isn’t really about single-source vs. mixed-source batching. I feel you did not understand my point about sampling.
>
> In your method, you’re simply masking out cross-modality pairs. but that’s exactly what proper batch sampling could achieve.
>
> For example, if your current batch looks like [T, I, T, T, V, V, I], we could just split it into three modality-homogeneous sub-batches: [T, T, T], [I, I], and [V, V], which would provide the same objective to your method.
>
> Since your method doesn’t allow any interaction across modalities anyway, computing one big masked batch is functionally identical to processing these three separate sub-batches. In that case, the masking serves no real purpose, it’s redundant engineering overhead with no methodological benefit.

---

> ### Author Response · Authors · 2025-12-03
> **Additional Response to Reviewer MRmf Part 4**
>
> Thank you for the correction. After re-checking the GTE series, we confirm that both GTE and mGTE explicitly state that they ensure “a batch originate from the same task”, while GME and GVE do not mention this detail. Given the GTE team’s consistently rigorous writing style, we assumed that had they applied the same strategy, they would have clearly specified it again. As the corresponding training code is unavailable, we cannot directly verify this behavior. We apologize if our earlier wording created any misleading impression.
>
> Regarding the discussion on “single-source vs. mixed-source batching”, this followed your initial comment on “prior contrastive learning works (e.g., sampling batches from a single data source to ensure uniform modality)”. We followed this terminology for consistency. Even so, as can be seen from our previous responses, our main focus is indeed on “modality”. We fully understand the strategy you describe, namely discretizing sampling along modality and constructing modality-homogeneous sub-batches. We agree that this is a promising idea for our scenarios. However, to the best of our knowledge, this approach currently lacks systematic empirical validation (for example, ablations and theoretical analysis) in multimodal retrieval. We therefore consider that this potential strategy is insufficient to invalidate the novelty or effectiveness of MAMCL.
>
> We have provided substantial evidence in support of our design. Influential work [5] reported instability training issues when single-source/modal batch sampling in multimodal settings. In addition, our “Additional Response to Reviewer MRmf Part 2” offers a view from gradient dynamics, explaining why MAMCL leads to more stable training. Our supplementary experiments also demonstrate that MAMCL is more effective.
>
> As for the idea of splitting each mixed batch into modality-homogeneous sub-batches, we agree this could be worth exploring, but its effectiveness for universal multimodal retrieval remains untested. Even if such a method turns out to work well, it does not conflict with our contribution. We would be very happy to see the community study this direction more deeply.
>
> We sincerely appreciate your active engagement in this discussion, which has helped us better understand related work and potential alternative strategies. In summary, regarding the concern that our masking approach may resemble GISTEmbed, we have clarified that MAMCL and GISTEmbed differ fundamentally in both motivation and methodology and are not directly comparable. As for whether the batch sampling strategy from GTE could similarly address cross-modal interference, we view it as an interesting and positive direction. However, the approach currently lacks empirical validation in multimodal retrieval, and whether it can serve as a substitute for MAMCL remains an open question.

---

### Official Review · Reviewer_oVez · 2025-10-30

**Soundness:** 4
**Presentation:** 3
**Contribution:** 3
**Rating:** 6
**Confidence:** 4

**Summary:**

This paper introduces UNITE, a multimodal embedding training paradigm in two stages -- retrieval adaptation followed by instruction tuning, and a loss variant called Modal-Aware Masked Contrastive Learning (MAMCL).
Specifically, MAMCL restricts negatives to the same target-modality type to reduce inter-modal interference.
The authors emphasize modality data curation (proportions and sequencing of TT/TI/TV pairs) and report state-of-the-art results on instruction-based and fine-grained retrieval benchmarks, notably MMEB and WebVid-CoVR.

**Strengths:**

1. Broad generalization and strong performance in video retrieval: the proposed UNITE models perform strong across various retrieval scenarios, tasks, and granularities. On WebVid-CoVR, UNITE_instruct-7B exceeds baselines under their reported settings.
2. Proper ablations: The paper includes a dedicated MAMCL ablation (Table 7) and a full training-data composition analysis (TT/TI/TV mix, under fixed data budget).

**Weaknesses:**

1. Marginal performance of the MAMCL component: while MAMCL is conceptually sound, its average gains are small (about +0.3 overall on MMEB, avg of +0.5 on WebVid-CoVR with 7B parameters), and it can trade off specific metrics (e.g., CoVR R@5 at 7B). I recommend deeper analysis on when/why it helps.
2. Lack of efficiency analysis: MAMCL changes the effective negative set via a modality mask, but the paper does not report compute comparisons to standard InfoNCE; only high-level training setup (e.g., 64×A100, single-epoch runs, time) is provided. Adding a theoretical analysis or a wall-clock comparison would strengthen this paper.

**Questions:**

See weaknesses

---

> ### Author Response · Authors · 2025-11-20
> **Response to Reviewer oVez**
>
> We appreciate the reviewer’s recognition of our work, including broad generalization, strong performance, and comprehensive ablation. The careful attention and constructive suggestions are invaluable. We have addressed the reviewer’s concerns in detail and improved our paper based on the feedback.
>
> ---
>
> ### Q1: Marginal performance of the MAMCL component. Recommend deeper analysis on when/why it helps.
>
> We sincerely appreciate the reviewer for careful attention to the MAMCL gains.
>
> We believe the improvements are conceptually important and demonstrate consistent effectiveness across diverse scenarios.
>
> 1. Consistency across scales, modalities, and tasks
>
> MAMCL shows stable positive gains across multiple model scales, modalities and tasks:
> - 2B model: +0.5 on MMEB overall, +1.7 on WebVid-CoVR (R@1), +1.1 average
> - 7B model: +0.3 on MMEB overall, +1.1 on WebVid-CoVR (R@1), +0.7 average
>
> 2. When and why MAMCL helps
>
> MAMCL is designed for multimodal, multi-task scenarios. Rather than optimizing for specific benchmarks, it ensures **cross-modal learning stability** in heterogeneous training setting by helping the model focus on relevant modality-specific features while avoiding noise from irrelevant negatives.
>
> For example, as shown in Tab. 7 (3→4), when MAMCL is not used, introducing MMEB fine-tuning leads to noticeable conflicts between MMEB and CoVR, resulting in a performance drop for CoVR (R@1 drops from 69.2 to 67.4). This occurs because CoVR constitutes only 15% of the mixed data, leading negatives from text/image-dominant tasks to introduce noise into video-dominant retrieval learning. However, after applying MAMCL (Tab. 7, 4→5), this issue is alleviated, recovering performance on CoVR R@1 (67.4→69.1).
>
> Our experimental setting encompasses 10+ input-output patterns (e.g., I→T, T→I+T, T+V→V) across 40+ retrieval tasks. The consistent improvements across this diverse landscape demonstrate MAMCL's effectiveness.
>
> Regarding the reviewer's concern about potential metric trade-offs (e.g., CoVR R@5 at 7B), we believe this performance fluctuation is a reasonable outcome of learning in complex multimodal and multi-task scenarios. These occasional performance dips do not affect the overall stable improvement in performance brought by MAMCL. To further demonstrate MAMCL's impact on fine-grained tasks, here are the performance improvements brought by MAMCL across various task categories:
>
> |Method|Classification (10 tasks)|VQA (10 tasks)|Retrieval (12 tasks)|Visual Grounding (4 tasks)|WebVid-CoVR R@1|
> |-|-|-|-|-|-|
> |2B w/o MAMCL|62.3|55.6|65.1|75.2|67.4|
> |2B w/  MAMCL|63.2 (+0.9)|55.9 (+0.3)|65.4 (+0.3)|75.6 (+0.4)|69.1 (+1.7)|
> |7B w/o MAMCL|68.6|65.0|70.6|84.1|71.4|
> |7B w/  MAMCL|68.3 (-0.3)|65.1 (+0.1)|71.6 (+1.0)|84.8 (+0.7)|72.5 (+1.1)|
>
> The results show that MAMCL brings consistent performance improvements across various tasks for both 2B and 7B models (except for the 7B Classification task). Overall, MAMCL is a method for training stability and cross-modal generalization, accompanied by performance improvements.
>
> ---
>
> ### Q2: Lack of efficiency analysis
>
> We sincerely appreciate the reviewer for this constructive suggestion.
>
> We agree that efficiency analysis is crucial for practical adoption. We provide both theoretical analysis and wall-clock measurements below, and commit to adding comprehensive results in the next version.
>
> 1. **Theoretical Complexity Analysis**
>
> Following the definitions in our paper, let $N$ be the batch size, $K$ be the number of negative candidates, and $D$ be the embedding dimension (for Unite-2B, $D=1536$, for Unite-7B, $D=3584$).
>
> For standard InfoNCE, the similarity matrix computation is: $O(N \cdot (1+K) \cdot D)$.
> For MAMCL, we only apply the mask before computing the similarity matrix: $O(N \cdot (1+K))$. The similarity matrix computation remains the same: $O(N \cdot (1+K) \cdot D)$.
> Thus, MAMCL only adds an extra computation of $O(N \cdot (1+K))$, which is **negligible compared to the similarity matrix computation** of $O(N \cdot (1+K) \cdot D)$ (since $D$ is sufficiently large).
>
> 2. **Actual Runtime**
>
> To further demonstrate the impact of MAMCL on training efficiency, below are the actual time costs (hours:minutes:seconds). The experimental setup is consistent with Tab. 13.
>
> |Method|2B|7B|
> |-|:-:|:-:|
> |w/o MAMCL|2:15:33|6:17:29|
> |w/ MAMCL|2:15:10|6:19:54|
> |Difference|-23s (-0.3%)|+2m25s (+0.6%)|
>
> The results show that the impact of MAMCL on training time is minimal, with the difference in training time being **less than 1%, which is negligible** (Even under identical training configurations, fluctuations of ~1% are common due to variations in GPU cluster and hardware scheduling). Therefore, we believe the computational cost of MAMCL does not introduce any significant overhead in practical training.
>
> ---
>
> We appreciate the reviewers' constructive suggestions. We will incorporate relevant supplementary into our paper to provide a more comprehensive coverage.

---

> > ### Comment · Reviewer_oVez · 2025-11-27
> >
> > Thank you for the detailed response. Your answers regarding the efficiency analysis were helpful, and I recommend adding the theoretical and wall-clock comparisons to the final manuscript to address the initial concerns.
> >
> > Regarding the performance contributions of MAMCL, the additional analysis demonstrating its role in stabilizing training within noisy settings are helpful and I would recommend adding this into the manuscript as well. However, I remain unconvinced that MAMCL represents a significant contribution to the field. The performance gains are marginal (e.g., overall 0.3% on MMEB for the 7B model). More importantly, the benefits appear to diminish as the model scales from 2B to 7B (e.g., performance gain +0.5% (2B) -> 0.3% (7B) on MMEB), and the authors have not provided other experiment to demonstrate MAMCL's potential to scale. This lack of a positive scaling trend raises concerns about the method's utility for larger foundation models.
> >
> > While the data curation findings are useful, the proposed algorithmic novelty (MAMCL) does not appear to be the primary driver of the reported SOTA results compared to the backbone and data engineering. Therefore, I maintain my score as 6.

---

### Official Review · Reviewer_bQSb · 2025-11-03

**Soundness:** 3
**Presentation:** 3
**Contribution:** 3
**Rating:** 6
**Confidence:** 4

**Summary:**

The paper proposes UNITE, a framework for building unified embeddings for multimodal information retrieval, supporting text, images, videos, and their combinations. A key contribution is the introduction of MAMCL (Masked Contrastive Learning), which mitigates interference by masking contrastive terms between candidates of different modality types, thereby improving cross-modal alignment in the shared embedding space. The training methodology is structured into two stages: a retrieval adaptation phase, which leverages diverse modality pairs to construct a robust embedding space, and an instruction tuning phase, where the model is fine-tuned on instruction-based datasets to handle more complex and nuanced retrieval queries. Experiments on multiple benchmarks show consistent improvements, and ablation studies confirm MAMCL’s effectiveness across various instruction-based retrieval tasks.

**Strengths:**

The paper introduces a novel Modality-Aware Masked Contrastive Learning (MAMCL) approach that extends contrastive learning to better accommodate heterogeneous modalities. The framework's ability to integrate video retrieval alongside text and image retrieval broadens its applicability and underscores the model's versatility in handling complex multimodal data.
Comprehensive evaluations across diverse multimodal benchmarks substantiate the benefits of both MAMCL and the modality-aware data design. The inclusion of both in-distribution and out-of-distribution analyses strengthens the empirical validity of the claims.
The paper is generally well-structured and clearly written. The motivation, methodology, and experimental results are communicated with precision, and the technical formulations are clearly presented and easy to follow. While some training details are appropriately included in the appendix, certain key aspects—such as the specific large language model (LLM) used and the preprocessing steps for each modality, particularly for video—would be better placed in the main body to improve transparency and reproducibility.
This work makes a notable contribution by proposing the MAMCL approach, which mitigates cross-modal interference and enhances alignment across heterogeneous modalities. The proposed framework advances efforts toward building universal multimodal embeddings that can jointly handle text, image, and video retrieval tasks. The results demonstrate clear improvements on several benchmarks, indicating the approach's practical value and potential for broader application.

**Weaknesses:**

While the paper presents a well-motivated and empirically supported approach, several aspects could be strengthened to enhance its clarity and overall impact.
1- Frozen projector and vision encoder:
The authors freeze the projector and vision encoder, but do not analyze the implications of this choice. It remains unclear how fine-tuning these components—particularly during instruction tuning—might affect multimodal alignment and retrieval performance. An ablation study comparing frozen versus trainable projectors would provide valuable insight into the trade-off between stability and adaptability, strengthening the paper’s empirical analysis.
2- Limited model diversity:
Extending experiments to other state-of-the-art vision–language models for multimodal information retrieval would provide more substantial evidence of robustness and generality, and help disentangle the contribution of MAMCL from the underlying model architecture.
3- Suboptimal video sampling strategy
The paper employs a uniform frame sampling rate of one frame per second for the video modality, which may be too coarse to capture meaningful temporal dynamics. This approach could overlook key motion cues or fine-grained visual changes that are crucial for accurate retrieval. Exploring more efficient or adaptive sampling strategies could improve video representation quality and strengthen overall multimodal retrieval performance.
4- Lack of qualitative success and failure examples:
The paper focuses heavily on quantitative results but omits illustrative examples of both successful and failed retrieval cases. Including such visual or textual samples would help readers better understand the model’s strengths, limitations, and failure patterns—particularly for instruction-based or fine-grained retrieval scenarios.

**Questions:**

- Have the authors considered conducting an ablation study with the projector unfrozen during training to examine its effect on learning dynamics, stability, and retrieval performance?
- Since MAMCL is presented as a general loss function, could the authors clarify why it was evaluated only on Qwen2-VL-2B?
Do you expect similar performance trends if integrated into other VLMs such as CLIP, BLIP-2, or LLaVA?
If computational limits prevented broader testing, could you provide reasoning or partial results indicating its generalizability?
- Have the authors considered testing alternative video sampling strategies (e.g., motion-based, content-aware, or adaptive sampling) instead of the fixed rate of one frame per second to evaluate their effect on retrieval accuracy and temporal representation quality?
- Would it be possible to include qualitative examples of both successful and failed retrieval cases (especially for instruction-based tasks)? Such examples could clarify where MAMCL contributes most effectively and where it struggles—insights that would be very helpful to the community.

---

> ### Author Response · Authors · 2025-11-20
> **Response to Reviewer bQSb Part 1**
>
> We thank the reviewers for recognizing the effectiveness of our proposed strategy. We truly appreciate the valuable time and feedback. We have carefully addressed all points that suggested additional exploration and we hope our clarifications relieve concerns. In addition, the suggestion to include qualitative cases is constructive. We have incorporated this into the paper and believe it will help readers gain a more intuitive understanding of the model’s effectiveness.
>
> ---
>
> ### Q1: Frozen projector and vision encoder
>
> We sincerely appreciate the reviewer for raising this comment.
>
> We would like to clarify the following points:
> 1. **Freezing the projector and vision encoder is a common practice**, consistent with influential works in the community (e.g., VLM2Vec, LamRA, GVE). This is not a unique aspect of our approach and does not impact the validity of our experimental results.
> 2. **The choice to freeze the vision module is not a core component of our method.** The core of our work lies in data curation and MAMCL, which provides insights into the combination of text, image, and video modalities and improvements across diverse complex tasks.
> We understand the suggestion to perform a comparison between frozen and trainable vision components, but this is orthogonal to the primary goals and contributions of our work. Whether or not the vision module is frozen does not affect our goals and does not lead to a fundamental change in our results or conclusions.
>
> ---
>
> ### Q2: Limited model diversity. Why it was evaluated only on Qwen2-VL-2B? Do you expect similar performance trends if integrated into other VLMs?
>
> We sincerely appreciate the reviewer for raising this important technical question.
>
> Our design choice is based on the following considerations and validation:
>
> 1. MAMCL is architecture-agnostic
>
> MAMCL is fundamentally a **pure mathematical modification** of the standard InfoNCE loss, where contrastive terms from non-target modalities are masked to reduce interference between candidates from different target modalities (Eq. 2-5), thereby improving cross-modal alignment performance. We expect MAMCL to be equally effective on other VLMs. However, integrating MAMCL into other VLMs requires framework adaptation and retraining on all data, which demands considerable computational resources beyond what we can provide during the discussion phase.
>
> 2. Qwen2-VL-2B for main experiments
>
> Our core contributions are data curation and MAMCL. We use Qwen2-VL architecture, aligning with many influential works (e.g., GME, UniME, CaRe) that use the same backbone. This eliminates architectural differences and focuses the contribution on the method itself. Regarding the choice of the 2B size, it follows the scaling law for LLMs, which is a common practice in the era of LLM: validate the effectiveness of the method on smaller models and then scale up to larger models. This is consistent with the approach in existing works (e.g., UniME, GME). In our main results and ablation studies, we have demonstrated the effectiveness of our approach. For example, in Tab. 1, Unite-2B achieved competitive results with the 7B baseline, and further improvements are observed on the 7B model. In Tab. 7, ablation analysis of both the 2B and 7B variants shows that MAMCL consistently leads to stable improvements across multiple tasks.
>
> We are optimistic about the generalizability of MAMCL across different VLMs. Given its simple yet effective modification, we expect similar results in other models. Our method's effectiveness is demonstrated through comparisons with task-specific models, fair comparisons with other Qwen2-VL-based frameworks, and experiments with our data curation and training strategies.
>
> ---
>
> ### Q3: Suboptimal video sampling strategy
>
> We sincerely appreciate the reviewer for this comment on the video sampling strategy.
>
> We would like to clarify two aspects:
> 1. **Uniform frame sampling is a consensus in the MLLM field.** 1 fps sampling is a widely adopted strategy in the video MLLM community, used in prominent works like LLaVA series and Qwen-VL series. Additionally, this universal sampling approach ensures a fair comparison with baselines, making it a completely reasonable choice.
> 2. **The focus of the study is modality curation, not video processing.** While novel video sampling methods are encouraged, they are not the primary focus of this work. Furthermore, more complex sampling methods may introduce additional computational overhead, potentially increasing training and inference time.
>
> If we use less common sampling strategies (e.g., motion-based or content-aware adaptive sampling), it could conflate the contribution of our method with the changes introduced by  sampling strategy, making it difficult to isolate the specific benefits of our contribution. Therefore, we follow the standard 1 fps sampling strategy to maintain the robustness of the experimental results.

---

> ### Author Response · Authors · 2025-11-20
> **Response to Reviewer bQSb Part 2**
>
> ### Q4: Lack of qualitative examples
>
> We sincerely appreciate the reviewer for this constructive suggestion.
>
> We agree that qualitative analysis enhances intuitive understanding of model behavior. We will adopt this suggestion and **add a Case Study section in the appendix** of the next version, presenting some retrieval cases, particularly for fine-grained and instruction-based retrieval scenarios (see Appendix D). Each case will be analyzed to help readers understand the model's performance in complex scenarios.
>
> ---
>
> We appreciate the reviewers' helpful suggestions for our paper, particularly the introduction of qualitative examples. We will update our paper to enable readers to better observe the results of our model across various tasks.

---

### Official Review · Reviewer_mqhb · 2025-11-05

**Soundness:** 3
**Presentation:** 2
**Contribution:** 3
**Rating:** 6
**Confidence:** 3

**Summary:**

The paper introduces a training strategy for large multimodal models (LMMs) for the task of cross-modal retrieval. The proposed strategy involves a masking matrix that informs the loss function about the type of data modality involved in the data used to calculate the contrastive objective. In addition, the paper conducts experiments that evaluate the importance of different data compositions for training, identifying data curation regimes that improve performance. The experimental results indicate that performance improves in various tasks and datasets.

**Strengths:**

* Comprehensive evaluation across datasets and tasks.
* Simple, yet effective strategy.
* Good experimental results that indicate effectiveness of the proposed strategy.
* Generally speaking the study seems well designed and well conducted.

**Weaknesses:**

* Writing style is too distracting. Every time the paper indicates that something is critical or crucial, it is not.
* The contribution seems incremental with tweaks to existing models and based primarily in data curation.
* The paper reports results that they say contradicts previous observations, but no reference results or citations are provided.
* The interpretation of results and the insights is limited to highlighting numeric differences.

**Questions:**

* Are the results really surprising? How can these observations be explained beyond performance differences? Any hypotheses that can be tested or just black box performance differences that cannot be explained?
* Are the data curation results just a selection of data useful to solve the benchmarks rather than learning generalizable reasoning to match content?

---

> ### Author Response · Authors · 2025-11-20
> **Response to Reviewer mqhb Part 1**
>
> We thank the reviewers for recognizing the strengths of our work, including effective strategy, comprehensive and convincing experimental results, and the sound design and implementation. We also appreciate the reviewers’ careful reading and professional feedback. We have thoroughly addressed the raised concerns, especially those regarding writing clarity and the generalizability of our method, and we are continuously updating the paper. We believe the reviewers’ valuable suggestions have improved the quality of our paper.
>
> ---
>
> ### Q1: Writing style is distracting. Every time the paper indicates that something is critical or crucial, it is not.
>
> We sincerely appreciate the reviewer's careful reading and feedback on our writing style.
>
> We acknowledge that some terms (e.g., "critical", "crucial") are used too heavily, which may have caused a distracting reading experience. We will revise these expressions in the next version to be more restrained. Specifically, for truly important points, such as key issues and core conclusions, we will retain the original expressions, while elsewhere (e.g., L373, L431, L437) we will replace them with milder terms (e.g., "practical").
>
> We will reassess these expressions and ensure clearer and more natural presentation in the next version.
>
> ---
>
> ### Q2: The contribution seems incremental with tweaks to existing models and based primarily in data curation.
>
> We sincerely appreciate the reviewer for raising this important concern regarding the scope and significance of our contributions.
>
> While data curation is indeed a core contribution, it is not the only one. Our contributions can be summarized into two parts:
> - **Data curation**: We provide the first systematic exploration of training data composition for unified text-image-video retrieval. Prior work lacks clear guidance on data selection and training paradigms for such complex scenarios. Our experiments (Tab. 8) analyze how different data types affect different tasks and reveal how to maximize data utilization during training.
> - **Modal-Aware Masked Contrastive Learning (MAMCL)**: Due to heterogeneity across modalities, indiscriminately mixing multimodal data during training leads to conflicts and suboptimal performance [1,2]. MAMCL addresses this issue, showing consistent improvements across multiple tasks (Tab. 7), enabling full utilization of multimodal data to match specialized models.
>
> The tasks we focus are challenging, involving multiple modalities, 10+ input-output patterns, and 40+ specific tasks. Previous works have not tackled such complexity. Our systematic approach addresses the challenges of handling heterogeneous data, providing empirical guidance and new insights for the community on efficiently leveraging multimodal, multi-task data.
>
> ---
>
> ### Q3: The paper reports results that they say contradicts previous observations, but no reference results or citations are provided.
>
> We sincerely appreciate the reviewer for the attention to the clarity of our claims.
>
> We apologize for the potential confusion in L397-405. We don't intend to contradict prior research findings, but rather to express that **our results contradict intuitive expectations**. Specifically, we find that using a mismatched data combination (e.g., training with TV data for a TI task) can outperform using the appropriate training data (e.g., TI data for TI tasks). Intuitively, one would expect TI data to perform better for TI tasks and TV data for TV tasks. However, our experiments show that training with TV data yields better results in both TI and TV test scenarios.
>
> We will revise this phrasing to: "Notably, in image-text retrieval tasks, training solely with TV data outperforms training that uses only TI data, which contradicts intuitive expectations in image-text retrieval."
>
> ---
>
> ### Q4: The interpretation of results and the insights is limited to highlighting numeric differences.
>
> We sincerely appreciate the reviewer for raising this insightful comment.
>
> While numeric differences provide direct validation of our method's effectiveness, we also provide qualitative insights (Sec. 4.1, 4.2, 5) that offer empirical guidance for future universal multimodal embedding research and theoretical methods for mitigating modality conflicts.
>
> We understand your concern and will supplement the revision with additional analysis and interpretation. Specifically, we will add visualized case studies in the appendix to enhance reader understanding.
>
> ---
>
> [1] Bridging Modalities: Improving Universal Multimodal Retrieval by Multimodal Large Language Models, CVPR 2025
> [2] Balanced Multimodal Learning: An Unidirectional Dynamic Interaction Perspective, arXiv:2509.02281

---

> ### Author Response · Authors · 2025-11-20
> **Response to Reviewer mqhb Part 2**
>
> ### Q5: Any hypotheses that can be tested or just black box performance differences that cannot be explained?
>
> We sincerely appreciate the reviewer for raising this thoughtful comment.
>
> Our core hypothesis is that strategic data curation and modality-aware learning configurations improve multimodal retrieval performance. This is supported by both quantitative experiments and qualitative analysis, grounded in the understanding that different modalities (text, image, video) have unique characteristics that may interfere with each other if not handled properly.
>
> Tab. 8 demonstrates how different data configurations perform across different tasks, while ablation studies (Tab. 7) validate MAMCL's effectiveness in mitigating modality conflicts. Therefore, our work is not merely black-box performance improvement, but rather provides an interpretable approach based on deep understanding of modality characteristics.
>
> ---
>
> ### Q6: Are the data curation results just a selection of data useful to solve the benchmarks rather than learning generalizable reasoning to match content?
>
> We sincerely appreciate the reviewer for raising this meaningful question on generalization.
>
> Our data curation strategy are not limited to optimizing specific benchmarks. Instead, we believe our data curation provides a generalizable, transferable method for universal multimodal retrieval. Specifically, through systematic experiments with different modality combinations, we find that:
> 1. Video-text pairs are more effective for cross-modal retrieval, outperforming image-text data even on image-text tasks, indicating that video-text provides higher-quality training signals.
> 2. Text-text pairs enhance linguistic understanding and reasoning, while image-text pairs provide precise multimodal alignment (more focused than video content, capturing key details). These are particularly beneficial for instruction-following retrieval scenarios.
>
> To further validate generalizability, we provide Unite's performance on various unseen benchmarks [3], including CameraBench (CMRB), DREAM-1K-Event (DREAM-E), LoVR-Theme2Clip (LoVR-TH), LoVR-Text2Video (LoVR-V), LoVR-Clip-to-Video (LoVR-C2V), PE-Video-Keyword (PEV-K), MSRVTT-ImageVideo (MSRVTT-I2V):
>
> | Method     | CMRB | DREAM-E | LoVR-TH | LoVR-V | LoVR-C2V | PEV-K | MSRVTT-I2V | Avg. |
> |-|:-:|:-:|:-:|:-:|:-:|:-:|:-:|:-:|
> |            | R@10 | R@1  | R@10 | R@1  | R@1  | R@1  | R@1  |      |
> | VLM2Vec-V2 | 28.6 | 22.8 | 49.2 | 61.0 | 38.5 | 32.4 | 84.1 | 45.2 |
> | GME-7B     | 30.4 | 27.4 | 52.3 | 71.0 | 37.0 | 39.6 | 86.0 | 49.1 |
> | Unite-7B   | 35.1 | 27.9 | 55.5 | 62.0 | 44.8 | 44.0 | 88.3 | 51.1 |
>
> Unite's strong performance across these unseen benchmarks demonstrates that **our approach learns generalizable multimodal matching capabilities rather than benchmark-specific optimizations**.
>
> ---
>
> [3] Towards Universal Video Retrieval: Generalizing Video Embedding via Synthesized Multimodal Pyramid Curriculum, arXiv:2510.27571
>
> ---
>
> We thank the reviewer again for the suggestions regarding our writing style and presentation. We have benefited greatly from the reviewers' suggestions, which have been very helpful in refining our paper.

---

### Author Response · Authors · 2025-12-03
**Summary of Rebuttal and Discussion**

Dear Area Chair,

We sincerely thank the reviewers for their thoughtful and constructive feedback. These comments have greatly enhanced the clarity and completeness of our work. We also appreciate the area chair for the time and consideration.

Below, we summarize the main points of consensus and explain how our rebuttal and supplementary experiments address the reviewers' concerns.

---

## Consensus on Strengths

1. **Effective strategy.** Reviewer mqhb considers our method simple yet effective. Reviewer bQSb notes that our approach has **broad applicability and strong generalization in complex multimodal scenarios**. Both bQSb and oVez note that the ablation studies on MAMCL confirm its effectiveness.
2. **Comprehensive evaluation.** All reviewers acknowledge that our evaluation is extensive: UNITE is tested on 40+ retrieval tasks and demonstrates **strong performance across diverse scenarios, tasks, and granularities**. UNITE outperforms several task-specific models and even larger competitors.
3. **Valuable analyses and insights.** All reviewers agree that our systematic analysis of different modality combinations provides **new findings and practical guidance** for multimodal retrieval data curation.

---

## Response to Main Concerns

1. **Presentation (mqhb).** We refined the terminology and phrasing throughout the paper and clarified several points. These revisions aim to enhance readability.
2. **Generalization (mqhb, MRmf)** Our approach is not tuned to specific benchmarks. To support this, we report additional results of UNITE on several unseen benchmarks, where it also achieves strong performance. These results confirm that our method is generalizable.
3. **Efficiency analysis (oVez)** We provided both a theoretical complexity analysis and actual runtime comparisons between standard InfoNCE and MAMCL. Both analyses show that MAMCL introduces negligible computational overhead (<1% fluctuation in practice).
4. **Qualitative analysis (mqhb, bQSb)** We added qualitative case studies (Appendix D). These examples cover fine-grained cross-modal retrieval and instruction-based retrieval, and illustrate that UNITE handles challenging examples well in practice.
5. **Model choices and data processing (bQSb).** We clarified that our choices regarding backbone selection, freezing partial parameters, and video sampling, follow standard practice in the community. Our core contributions are orthogonal to these choices and remain valid even if alternative configurations are used.
6. **Theoretical and modal coverage in modality curation (MRmf)** We provided theoretical reasoning and  examples to explain our findings. We clarified that the original paper already analyzes the distinct roles of T–T, T–I, and T–V, and in the rebuttal we refined this analysis with clearer tables and discussions. We further explained how these findings directly informed our final data composition (proportions of T–T, T–I, T–V).
7. **Deeper analysis of MAMCL (oVez, MRmf).** For reviewer oVez, we presented additional analyses showing that MAMCL brings consistent improvements across model scales, modalities, and task types, and stabilizes training in noisy data settings. For reviewer MRmf’s concerns about novelty, we carefully compared MAMCL with prior masking (GISTEmbed) and batching strategies (GTE), highlighting conceptual and mechanistic differences. We combined community practice, influential recent work, analysis from the gradient dynamics, and additional experiments to clarify that MAMCL is both well motivated and empirically effective in heterogeneous multimodal retrieval.

---

we believe our response and additional experiments have addressed the reviewers’ main concerns and further validated the effectiveness, robustness, and generalization of our approach.

We sincerely thank the reviewers and Area Chair for the time and efforts.

Best regards,
Authors

---

### Meta-Review · Area_Chair_rBJx · 2025-12-30

**Summary:**

This paper received a recommendation score of 6, 6, 6, 4 after rebuttal, meaning three reviewers gave positive scores, while one reviewer maintained a negative score after rebuttal. Based on the reviews and the rebuttal, the paper has the following problems: 1) Limited novelty; 2) Reviewers found that the reported state-of-the-art performance appears to be driven more by data curation and engineering choices than by this methodological innovation; 3) The paper lacks sufficient theoretical explanation for its findings, deeper analysis of its ablation studies, and critical implementation details for reproducibility. Considering these issues, the paper was rejected.

**Reviewer Concerns:**

Some of the concerns have been addressed, but the key concern regarding the novelty proposed by Reviewer MRmf still remains.

**Reviewer Scores:**

There is no clear indication that the reviewers might increase their scores. The final scores are 6, 6, 6, 4.

---

### Decision · Program_Chairs · 2026-01-26

Reject